# On the Existence and Complexity of Core-Stable Data Exchanges

**Jiaxin Song**
University of Illinois, Urbana-Champaign
jiaxins8@illinois.edu

**Pooja Ravi Kulkarni**
Northwestern University
pooja.kulkarni@northwestern.edu

**Parnian Shahkar**
University of California, Irvine
shahkarp@uci.edu

**Bhaskar Ray Chaudhury**
University of Illinois, Urbana-Champaign
braycha@illinois.edu

## Abstract

The rapid growth of data-driven technologies and the emergence of various data-sharing paradigms have underscored the need for efficient and stable data exchange protocols. In any such exchange, agents must carefully balance the benefit of acquiring valuable data against the cost of sharing their own. Ensuring stability in these exchanges is essential to prevent agents—or groups of agents—from departing and conducting local (and potentially more favorable) exchanges among themselves. To address this, we study a model where $n$ agents participate in a data exchange. Each agent has an associated payoff for the data acquired from other agents and a cost incurred during sharing its own data. The net utility of an agent is payoff minus the cost. We adapt the classical notion of *core-stability* from cooperative game theory to data exchange. A data exchange is core-stable if no subset of agents has any incentive to deviate to a different exchange. We show that a core-stable data exchange is guaranteed to exist when agents have concave payoff functions and convex cost functions– a setting typical in domains like PAC learning and random discovery models. We show that relaxing either of the foregoing conditions may result in the nonexistence of core-stable data exchanges. Then, we prove that finding a core-stable exchange is PPAD-*hard*, even when the potential blocking coalitions are restricted to constant size. This provides the first known PPAD-hardness result for core-like guarantees in data economics [BGI$^+$24a, ACGM25]. Finally, we show that data exchange can be modeled as a *balanced* $n$-person game. This immediately gives a pivoting algorithm via Scarf's theorem [Sca67]. We show that the pivoting algorithm works well in practice through our empirical results.

## 1 Introduction

From accelerating vaccine development in healthcare to improving fraud detection in financial services and advancing self-driving technology in the automotive industry, *high-quality data* has become the bedrock of algorithmic decision-making and AI-driven solutions in the 21st century. However, this valuable data is often fragmented i.e., distributed across multiple organizations. This decentralization makes data sharing and collaboration critical for effective decision-making. For instance, in healthcare, capturing nuanced relationships between disease patterns, socio-economic factors, genetic information, and rare conditions requires training machine learning models on large, diverse datasets that extend beyond curated datasets within individual organizations [RHL$^+$20]. Similarly, in autonomous vehicle technology, self-driving cars must be exposed to a wide range

of driving scenarios, including varying terrains, climates, and traffic regulations across countries, making it essential to train models on diverse datasets from multiple regions [XPHS23]. Collaborative data economics offers both unique opportunities and challenges: Unlike most economic assets, data is *non-rivalrous*– meaning multiple organizations can simultaneously benefit from the same data. This creates the potential for significantly greater collective value compared to traditional market economies, where assets are typically rivalrous. However, the benefits of data sharing are tempered by concerns over privacy, security, and the risk of losing a competitive advantage. As a result, despite its vast potential, data collaboration has not yet reached its full scale. As the European Council aptly notes [Com20]:

"*In spite of the economic potential, data sharing between companies has not taken off at a sufficient scale. This is due to a lack of economic incentives, including the fear of losing a competitive edge.*"

We introduce a collaborative data exchange economy, where a group of agents, each having an endowment of data, aim to engage in mutually beneficial data exchanges. Each agent derives (i) a payoff from the data acquired (indicative of the agent's value for the marginal improvement in predictive payoff of their ML model from the acquired data) and (ii) incurs a cost for sharing their own data. The agent's net utility from the data exchange is defined as the payoff minus the cost. In our model, shared data cannot be redistributed by other agents. Specifically, when we say that agent $i$ shares its data with agent $j$, we mean that agent $i$ allows agent $j$ to refine its machine learning model using samples of $i$'s data. However, this does not entail direct data sharing between $i$ and $j$. Instead, we adopt a framework similar to standard collaborative learning paradigms, such as Federated Learning [MMR+17], where $j$ trains on $i$'s data by only receiving the gradients of the loss function on $i$'s data, allowing agent $j$ to update its model without direct access to $i$'s data.

**Desiderata.** The gold-standard desiderata in a collaborative economy is *core-stability*– a data exchange is core-stable if no group of agents can identify a local exchange among themselves that they all strictly prefer to the current exchange. Core-stability implies other desired guarantees like (i) *individual rationality:* every agent participating in the data exchange gains more in utility than they lose in the cost of sharing their own data, and (ii) *Pareto-optimality:* there exists no exchange that all agents strictly prefer to the current exchange. This paper delves into the conditions under which core-stable exchanges exist within this collaborative data exchange economy and explores how to compute such exchanges.

## 1.1 Our Contributions

**Data Exchange Model.** In an instance of our problem, there are $n$ agents $N$, where each agent $i$ owns dataset $D_i$. In a data exchange $\boldsymbol{x} = (x_{i,j})_{i \neq j \in [n]}$, $x_{i,j} \in [0, 1]$ represents the fraction of $D_i$ shared with $j$, $x_{-i}$ represents the data bundle $(x_{1,i}, x_{2,i}, \ldots, x_{n,i})$ that $i$ receives from the data exchange, and $x_i$ represents the data bundle $(x_{i,1}, x_{i,2}, \ldots, x_{i,n})$ that $i$ gives to the data exchange. Given a data exchange $x$, an agent's utility $u_i(\boldsymbol{x})$ is given by

$$u_i(\boldsymbol{x}) = p_i(x_{-i}) - c_i(x_i),$$

where $p_i(x_{-i}) \in \mathbb{R}_{\geq 0}$ and $c_i(x_i) \in \mathbb{R}_{\geq 0}$ denote the *payoff* agent $i$ has from $x_{-i}$ and the *cost* incurred by sharing $x_i$ for agent $i$ respectively. We assume $p_i(\boldsymbol{0}) = c_i(\boldsymbol{0}) = 0$, which means that agent $i$ receives no benefit and suffers no cost if she is not exchanging any data with others. Further, consistent with the existing literature on data-sharing [MYC+23, KGJ22, BHPS21a], $p_i(\cdot)$ and $c_i(\cdot)$ are *monotone* in $x_{ji}$ and $x_{ij}$ for all $j$, i.e., more data acquired gives higher payoff, and more data shared leads to higher costs.

*Core-Stability.* A data exchange $x$ is core-stable if there exists no coalition of agents $U \subseteq N$, and an exchange $\boldsymbol{x}^U$ among agents in $U$ such that $u_i(\boldsymbol{x}^U) > u_i(\boldsymbol{x})$ for all $i \in U$.

**On the Existence of Core-Stable Exchanges.** Our first observation is that a core-stable data exchange may not always exist, even when there are only three agents. This motivates studying natural conditions that guarantee the existence of a core-stable data exchange. To this end, we prove that when agents have concave payoff functions and convex cost functions, a core-stable data exchange always exists. The foregoing conditions capture a broad range of interesting instances [BHPS21a, KGJ22]. Convexity of cost is a natural choice since it captures the property of increasing marginal costs. For

instance, data sharing through *ordered selection*, i.e., sharing records in ascending order of costs involved for collecting the records, results in convex cost functions. There are more models that result in strictly convex cost functions ([LR14]). Similarly, several important ML models exhibit concave payoff functions; for instance, payoffs in linear or random discovery models [BHPS21b], random coverage models [BHPS21b], and general PAC learning [MRT18] are all concave. Furthermore, there is empirical evidence that the accuracy function in neural networks under the cross-entropy loss is also concave ([KMH$^+$20]). We further show that relaxing either of these conditions, i.e., concavity of payoff or convexity of cost, can lead to instances that do not admit core-stable exchanges.

**Theorem 1.** *When agents have concave payoff functions and convex cost functions, a core-stable data exchange always exists. One can construct instances relaxing only one of the foregoing conditions (either concavity in payoff or convexity in costs) that do not admit any core-stable data exchanges.*

**Computational Results.** We allow oracle access to the value and super gradient of the utility functions. In particular, we have the following two types of queries: *Value query $\mathcal{U}(i, \boldsymbol{x})$*: return the utility that agent $i$ receives from the data exchange $\boldsymbol{x}$, $u_i(\boldsymbol{x})$; *Supergradident query $\nabla \mathcal{U}(i, \boldsymbol{x})$*: return the supergradident of utility function $u_i(\boldsymbol{x})$ at $\boldsymbol{x}$ if $u_i(\cdot)$ is concave.

We first prove that given an arbitrary instance, determining whether it admits a core-stable data exchange is NP-hard. Thereafter, we investigate the computational complexity of identifying core-stable exchanges under sufficient conditions. We show that finding a core-stable exchange under our sufficient conditions is PPAD-hard. Even if we restrict the blocking coalitions to comprise of only constantly many agents, the problem remains PPAD-hard. To the best of our knowledge, this is the first PPAD-hardness proof for core-like guarantees in data economies [BGI$^+$24a, ACGM25]. Our proof technique could be potentially useful for settling the complexity of the problems in [BGI$^+$24a, ACGM25].

**Theorem 2.** *Determining core-stable data exchanges when agents have concave monotone payoff functions and convex monotone cost functions is PPAD-hard. The hardness holds even when we restrict ourselves to our deviating coalitions of constant size. Further, for instances exhibiting non-concave payoffs and non-convex costs, it is NP-hard to determine whether a core-stable data exchange exists.*

On the positive side, our existence proof yields a pivoting algorithm to find a $\varepsilon$-approximate core-stable data exchange. Typically, pivoting algorithms (Simplex [Dan90], Complementary Pivot Algorithms [CDRP08]) are well-suited for practical implementation, suggesting that the problem may have effective algorithmic solutions in practice, despite the PPAD-hardness. In Section 4, we validate the practical efficacy of our algorithm through simulations on a mean estimation task similar to [BGI$^+$24a]. In particular, we observe that the number of "pivoting" operations, which in theory may not be polynomially bounded, grows linearly with the number of agents.

## 1.2 Related Work

Our work draws on concepts, techniques, and problems from several disciplines, including cooperative game theory, game complexity, and data economics. Providing a comprehensive survey of all related work is beyond the scope of this paper. Instead, we focus on (i) federated learning– a parallel framework to ours which also involves incentives and other economic processes involving data as an asset, and (ii) some related stability problems in cooperative games and their complexity.

**Federated Learning.** Federated learning (FL) offers a privacy-preserving and effective distributed learning paradigm in which a group of agents with local data samples collaboratively train a shared machine learning model [MMR$^+$17]. This approach has seen widespread success in applications like autonomous vehicles [ESC20] and digital healthcare [DRZ$^+$21, XGS$^+$21]. Data exchange can be viewed as a private learning paradigm, where each agent exchanges its own data for other valuable data to train its own private ML model. In contrast, FL is a public learning paradigm, where all agents collaboratively train the same model using data shared among them. Despite these differences, both FL and data exchange face similar challenges in designing principles and mechanisms to incentivize participation. As a result, FL has incorporated a range of concepts from game theory, including Stackelberg games [KPT$^+$20, PTB$^+$19], non-cooperative games [ZFX$^+$19, CZXL21], auctions [RSR$^+$21], incentives in collaboration [ADNMM25, PSS25], and budget-balanced reward mechanisms [MYC$^+$23, MSS$^+$25].

**Data Markets.** Data markets are two-sided real-time platforms facilitating pricing and selling data to data-seekers. Theoretical research on data markets has recently gained traction, given the current importance of data economies. There is a long line of work [AP86, AP90, BBS18, BKL12] that investigates revenue-maximizing strategies of a monopolist data seller. In fact, several studies investigate the pricing of data/ information from first principles in different settings [MDJM21, Pei20, CV20, BBG22]. Competitive pricing and allocation rules have also been discussed in the context of digital goods that behave similarly to data [JV10]. There is another line of theoretical work studying the data exchange economy [BGI+24a, ACGM25]. The setting of [ACGM25] is the most relevant one to ours, and they also consider the same data sharing model. In contrast to our work, [ACGM25] assumes no cost within data sharing, which makes it trivial to obtain a core-stable data exchange in their setting – everyone exchanges all the data. This way, no agent can achieve a higher utility in any exchange. Therefore, [ACGM25] investigates core stability in conjunction with another desideratum, i.e., fairness, and proves their existence results. The utility function of our setting is a generalization – an agent's utility is defined as payoff minus cost. As a result, the overall utility may not be non-monotonic, which introduces significant complexity. In particular, [ACGM25]'s results do not carry over, as we demonstrate in Example 1, a core-stable exchange may not even exist under the generality of [ACGM25]'s model when such costs are taken into account. This motivates our investigation into sufficient conditions – such as concave payoffs and convex costs – that still capture important and realistic scenarios while allowing for meaningful analysis. Conceptually, our utility model aligns more closely with practical distributed learning frameworks such as federated learning, where agents must consider both the benefits and costs of participation [MYC+23, MSS+25, PSS25].

**Cooperative Games and their Complexity.** Cooperative games focus on analyzing mechanisms and studying stable configurations in environments where agents voluntarily cooperate, in contrast to non-cooperative games where agents act independently and selfishly. Similar to *Nash Equilibrium* in non-cooperative games, *core-stability* is the canonical stability notion in cooperative games. The existence of core-stability has been investigated thoroughly within *transerable utility* (Bondareva-Shapley theorem [Sha67, Bon63]) and non-transferable utility cooperative games (Scarf's Theorem [Sca67]). Scarf's theorem has since been used to show the existence of stability in other cooperative settings like stable marriages [FHS23], fractional dominating antichains [AF03], and fractional stable hypergraph matching [AF03]. [Kin08] showed that computing the core of an $n$ person game (outcome of the Scarf's theorem) is PPAD-complete. The hardness of Scarf has then been used to show the hardness of several other cooperative problems (see [KPR+09] for a detailed outline).

**Convex Cost Functions.** Theorem 3 shows the existence of a core-stable data exchange under convex cost functions. Such cost functions have been discussed in the literature [LR14]. As discussed by [LR14, Section 3.1], when data collectors (e.g., grocery store) collect (private) data from agents (e.g., buyers) in return for financial compensation (e.g., membership card), the resulting marginal costs grow with the amount of data collected, which naturally leads to convexity. For example, early records are cheaper or less sensitive, while later records may carry greater privacy risks or legal implications. Similar convex models are also adopted in incentive and fairness studies for federated learning (e.g., [MYC+23, MSS+25, KGJ22]).

## 2 Existence of Core-Stable Data Exchanges

**Basic Notations and Definitions.** A *coalition* is a non-empty subset of the agents and a *deviation* in a data exchange $x$ is a pair $(U, x^U)$, where $U$ is a subset of agents and $x^U$ is a data exchange within $U$, i.e., $x_{i,j}^U > 0$ only if $i, j \in U$. A deviation *blocks* the data exchange $x$ if $u_i(x^U) > u_i(x)$ for all $i \in U$. An exchange $x$ is *core-stable* if there does not exist a deviation that blocks $x$. Formally,

**Definition 1** (Core-Stability and Relaxations). A data exchange $x$ is *core-stable* if for any $U \subseteq [n]$, there does not exist an exchange $x^U$ over $U$ such that $u_i(x^U) > u_i(x)$ for all $i \in U$. Further, $x$ is *core-stable with respect to s* if the above constraint holds for all $U \subseteq [n]$ with $|U| \leq s$. $x$ is $\alpha$-*core-stable* if for any $U \subseteq [n]$, there does not exist an exchange $x^U$ over $U$ such that $u_i(x^U) > u_i(x) + \alpha$ for all $i \in U$.

In this section, we provide a complete picture on the existence of core-stable data exchanges. In Section 2.1, we show that a core-stable data exchange may not always exist, even for the case of three agents. Despite this, we show that a core-stable data exchange always exists in Section 2.2, when (i)

the payoff functions are concave and (ii) the cost functions are convex. From here on, we refer to the foregoing two conditions as *sufficient conditions*. We also show that core-stable data exchange may not always exist if one of the two sufficient conditions is unsatisfied.

## 2.1 Non-existence of Core-Stable Data Exchange

We first give the reader the main idea of why a core-stable data exchange may not always exist.

**Example 1.** *Consider a data exchange instance with three agents, as illustrated in Figure 1, where the nodes represent the agents. For any agent $i \in [3]$, the green number (denoted by $p_{j,i}$) on the incoming edge $(j, i)$ denotes the payoff agent $i$ receives if agent $j$ shares her full dataset, while the red number (denoted by $c_{i,j}$) on the outgoing edge $(i, j)$ indicates the cost incurred by agent $i$ for sharing her full dataset with agent $j$. For example, as shown in Fig. 1, when agent $1$ shares her entire dataset with agent $2$, agent $2$ receives a payoff of $1/4$ while agent $1$ incurs a cost of $1/4$.*

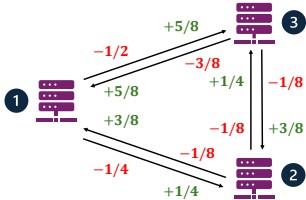

Figure 1: Payoff and cost for one-unit sharing, where a numbered node represents an agent and the arrow $i \to j$ represents that agent $i$ is sharing her entrie dataset with agent $j$, i.e., $x_{i,j} = 1$. The green number on the arrow means the payoff that agent $j$ receives by the single share, and the red number means the cost that agent $i$ incurs.

*For any agent $i \in [3]$, define her payoff and cost functions as follows.*

▷ *Payoff:*
$$p_i(x_{\alpha,i}, x_{\beta,i}) = \frac{p_{\alpha,i}}{\epsilon} \cdot (x_{\alpha,i} - (1 - \epsilon))_+ + \frac{p_{\beta,i}}{\epsilon} \cdot (x_{\beta,i} - (1 - \epsilon))_+ , \quad (1)$$

▷ *Cost:*
$$c_i(x_{i,\alpha}, x_{i,\beta}) = c_{i,\alpha} \cdot x_{i,\alpha} + c_{i,\beta} \cdot x_{i,\beta} + \frac{1}{\epsilon} \cdot x_{i,\alpha} \cdot x_{i,\beta}, \quad (2)$$

*where $\{\alpha, \beta\} = [3] \setminus \{i\}$ and $(\cdot)_+ = \max(\cdot, 0)$. The idea behind the payoff is agent $i$ benefits from agent $j$'s share only when $x_{j,i} > 1 - \epsilon$. Meanwhile, the cost function discourages an agent from sharing a fraction in the range $(0, 1 - \epsilon]$, as it does not yield a positive payoff for the other agent and only incurs a cost. In addition, no agent can share data with two other agents without incurring a high cost (which negates the benefit of getting any amount of data).*

*The values $p_{\alpha,i}, p_{\beta,i}, c_{i,\alpha}, c_{i,\beta}$ are curated carefully so that (1) If nobody shares anything, a couple of agents deviate and start sharing data with one another. (2) If only two agents exchange data, there is always one agent who prefers sharing with the third non-included agent, therefore she deviates with the non-included agent. (3) In a cyclic data exchange among the three agents, there always exists an agent for whom the cost of sharing data is more than the gain of receiving data, therefore, she prefers to deviate and have no data exchange. Therefore, for any data exchange, there exists a subset of agents that gain a strictly higher utility by deviation, implying a core stable exchange does not exist. We refer the reader to Appendix A.3 for further details, and a complete landscape of non-existence scenarios.*

## 2.2 Existence of Core-stable Data Exchange under Sufficient Conditions

Despite the non-existence in the general setting, we next show that core-stable data exchanges exist for a broad class of interesting instances that exhibit concave payoff functions and convex cost functions. To prove existence, we formulate the data exchange problem as an $n$-person game (proposed by [Sca67]) and then show that the game is balanced, which implies that a core exists.

*An $n$-person game* consists of $n$ agents and a function $V(\cdot)$. For every subset $S$ of $N$, function $V(S)$ specifies the set of outcomes $V(S)$, which consists of a set of utility vectors that are achievable by the coalition $S$. A utility vector $\boldsymbol{u}$ is *attainable* by $S$ if $\boldsymbol{u} \in V(S)$. The game is *balanced* if for any collection of coalitions $T$ along with a collection of nonnegative weights $\{\delta_S\}_{S \in T}$, if $\sum_{S|i \in S} \delta_S = 1$ holds for all $i \in N$, then utility vector $\boldsymbol{u}$ is attainable by $N$ if $\boldsymbol{u}_S$ is attainable by $S$ for all $S \in T$.

Next, we formulate the data exchange problem into the framework of the $n$-person game. Define $V(\cdot)$ as follows: for every subset $S \subseteq N$, $V(S)$ is the set of nonnegative utility vectors when only

agents in $S$ exchanging data. As the utility function is continuous and achieves zero when no data is exchanged, we demonstrate that $V(S)$ is nonempty and also downward closed (i.e., if $\boldsymbol{u} \in V(S)$ and $\boldsymbol{u}' \le \boldsymbol{u}$, then $\boldsymbol{u}' \in V(S)$) for any subset $S \subseteq N$. In addition, for any collection of coalitions $T$, weights $\{\delta_S : S \in T\}$ and utility vector $\boldsymbol{u}$ that meet the precondition of the balanced game, assume the utility vector $\boldsymbol{u}_S$ is achieved by data exchange $\boldsymbol{x}_S$. Consider the combination of the data exchanges $\boldsymbol{x} = \sum_{S \in T} \delta_S \cdot \boldsymbol{x}_S$. As $u_i(\cdot)$ is concave, for any $i \in N$, we have

$$u_i(\boldsymbol{x}) = u_i \left( \sum_{S \in T} \delta_S \cdot \boldsymbol{x}_S \right) \ge \sum_{S \in T: i \in S} \delta_S \cdot u_i(\boldsymbol{x}_S) = \sum_{S \in T: i \in S} \delta_S \cdot \boldsymbol{u}_i = \boldsymbol{u}_i,$$

As $V(N)$ is nonempty and downward closed, then utility vector $\boldsymbol{u}$ is attainable by $N$. Therefore, the data exchange game is balanced and the core exists. We defer the complete proof to Appendix A.2.

**Theorem 3.** *A core-stable data exchange always exists if the payoff functions $\{p_i\}_{i \in N}$ are concave, the cost functions $\{c_i\}_{i \in N}$ are convex.*

We remark that the above existence also extends to non-monotone payoffs and costs as long as they are concave and convex, respectively.

## 3 The Complexity of Finding Core-Stable Data Exchanges

In this section, we explore the computational complexity of identifying core-stable data exchanges. We first show that for instances that do not meet our sufficient conditions, determining the existence of a core-stable data exchange is NP-hard. Next, we shift our focus to finding core-stable data exchanges for instances that satisfy sufficient conditions and establish that this problem is PPAD-hard. We defer the full details of the NP-hardness proof to the Appendix and give an overview of the PPAD-hardness, which constitutes our main technical result.

**Theorem 4.** *It is NP-hard to determine the existence of a core-stable data exchange.*

### 3.1 PPAD-**Hardness of Finding Core-Stable Data Exchange**

We perform a reduction from the Approximate Fractional Hypergraph Matching problem which is known to be PPAD-hard [IK18, Csá22].

**Definition 2** (Approximate Fractional Hypergraph Matching Problem). In an instance of the fractional hypergraph matching, we are given a hypergraph $G = (V, E)$ and a preference order $\succ_v$ over hyperedges $E$ incident to $v$ for every vertex $v \in V$. Denote by $e' \succeq_v e$ if $e' \succ_v e$ or $e' = e$. Let $E(v)$ denote the set of hyperedges that contain/are incident to $v$. A fractional matching $f$ in $G$, assigns to each edge $e \in E$, a value $f(e) \in [0,1]$ such that $\sum_{e:\, e \in E(v)} f(e) \le 1$ for all $v \in V$.

**Definition 3** (Stable fractional matching). A fractional matching $f$ is $(1-\epsilon)$-*stable* if for every edge $e$, there exists a vertex $v$ in $e$ such that $\sum_{e' \in E(v):\, e' \succeq_v e} f(e') \ge 1 - \epsilon$.

Finding an approximately stable fractional matching remains PPAD-hard for any $\epsilon \le 1/2^{20|V|^4}$, even in 3-uniform hypergraphs with maximum vertex degree three [Csá22, IK18]. We reduce the problem of finding an approximately stable fractional matching to the problem of finding a core-stable data exchange. We give the detailed proof in Appendix B.2.1 of the Appendix, but include a shorter overview of the reduction in the main body.

**Theorem 2.** *Determining core-stable data exchanges when agents have concave monotone payoff functions and convex monotone cost functions is PPAD-hard. The hardness holds even when we restrict ourselves to our deviating coalitions of constant size. Further, for instances exhibiting non-concave payoffs and non-convex costs, it is NP-hard to determine whether a core-stable data exchange exists.*

**Overview of the Reduction.** To see the connection to our problem, we urge the reader to interpret the vertices of $G$ as agents, the hyperedges $E$ as coalitions, and the matching $f$ as a function that assigns to each edge $e$ (coalition) a value. The stability criterion in fractional hypergraph matching requires that for each hyperedge (coalition) $e$, there is at least one vertex $v$ incident to $e$ (one agent that is part of the coalition $e$) such that the total aggregated value on edges preferred strictly or equal to $e$ by $v$ is large: $\sum_{e' \in E(v): e' \succeq_v e} f(e') \ge 1 - \epsilon$.

Given an instance of fractional hypergraph matching, we construct a data exchange instance, where we have agent $v_a$ corresponding to a vertex $v$ in $G$. Further, each blocking coalition corresponds to a hyperedge. We then show that if the core-stability condition holds in the data exchange problem — i.e., for every blocking coalition corresponding to a hyperedge $e$, and any exchange within that coalition, there exists an agent $v_a$ who prefers the current exchange — then for the corresponding hyperedge $e$ in $G$, we have $\sum_{e' \in E(v): e' \succeq_v e} f(e') \geq 1 - \epsilon$ for the agent $v$.

The data exchange instance is constructed in the following way: Given an instance $(G = (V, E), \{\succ_v\}_{v \in V})$ of fractional hypergraph matching, with $|E(v)| \leq 3$ and every hyperedge containing three vertices (if one hyperedge has less than three vertices, we can add dummy vertices to it), we construct an edge-agent $e_a$ for every edge $e$, and a vertex-agent $v_a$ for every vertex $v$. Every vertex agent $v_a$ is only interested in (has non-zero marginal utility for) the data of the edge agents corresponding to the hyperedges incident to $v$, and every edge agent $e_a$ is only interested in the data hosted by the vertex agents corresponding to the vertices incident to $e$. Moreover, for every $\Delta > 0$, every edge agent $e_a$ is willing to exchange $\Delta$ units of its data with the vertex agents corresponding to the vertices in $e$, for $\Delta$ units of data from them, i.e., the payoff gain from receiving $\Delta$ units of data from the vertex agents compensates the cost of sharing $\Delta$ units of data with them. The payoff of a vertex agent $v_a$ is defined as $\sum_{e \in E(v)} w(v_a, e_a) x_{e_a v_a}$ where $w(v_a, e_a)$ is the utility agent $v_a$ gets from unit data of $e_a$. We set the weights $(w(\cdot, \cdot))$ such that $e \succ_v e'$, implies $w(v_a, e_a) > w(v_a, e'_a)$. The cost of any vertex agent $v_a$, $c_{v_a}(x)$ is $\Gamma \cdot (\sum_{e: v \in e} x_{v_a, e_a} - 1)_+$[1], for a sufficiently large $\Gamma$, which ensures that $\sum_{e \in E(v)} x_{v_a, e_a} \leq 1 + \gamma$ for a sufficiently small $\gamma > 0$, as otherwise agent $v_a$ will have negative utility and data exchange $\boldsymbol{x}$ will not be individually rational, and consequently not core-stable.

**Core Stability $\Rightarrow$ Stable Matching.** Now we show that ensuring core-stability in data exchange implies stability in fractional hypergraph matching. To this end, first observe that in fractional hypergraph matching, the stability criterion involves one variable per hyperedge, which appears in the inequality associated with every vertex incident to that hyperedge (See Definition 3). However, for core-stability in data exchange, inequality in Definition 1 involves distinct variables (in particular variables $x_{e_a, v_a}, x_{v_a, e_a}$ for every vertex $v$ incident to $e$, as the payoff and cost functions of $v_a$ are functions of these variables). To overcome the foregoing dilemma, we introduce more agents (call them *intermediate agents*), and carefully design their cost and payoff functions, such that if $x$ is a core-stable data exchange, then $x_{e_a, v_a} = x_{e_a, v'_a} \approx x_{v_a, e_a} = x_{v'_a, e_a}$ for all $v, v'$ incident to $e$. In particular, for each edge $e$, we introduce a set of intermediate vertices $I_e = \{i_e \mid e \in E(v)\}$ such that each $i_e$ acts as an intermediary between $e_a$ and $v_a$. The payoff and cost functions of the intermediate agents are designed in such a way that we ensure the exchanges between intermediary vertices with their corresponding vertex agents and edge agents are almost the same(See Figure 6). We refer the reader to Appendix B.2.2 for full details. Still, for the remainder of this Section, we proceed assuming that in a core-stable data exchange, all pairwise data exchanges between an edge agent and its corresponding vertex agents have the same value. We set $f(e)$ to this value.

**Lemma 1.** *If data exchange $\boldsymbol{x}$ is core-stable, then the fractional matching $f$ is $(1 - \epsilon)$-stable.*

*Proof.* We first show that $f$ is a valid fractional matching: Recall that by the design of the cost functions of the vertex agents, we ensure that $\sum_{e \in E(v)} x_{v_a, e_a} \leq 1 + \gamma$ for all $v$. Since we ensure that $x_{v_a, e_a} = x_{e_a, v_a} = f(e)$, we have $\sum_{e \in E(v)} f(e) \leq 1 + \gamma \leq 1 + \epsilon$ for a sufficiently small $\gamma$. Therefore, $f$ is a fractional matching.

We next show that $f$ also satisfies the stability criterion. Observe that the payoff of any vertex agent $v_a$ is $\sum_{e \in E(v)} w(e_a, v_a) x_{e_a, v_a}$. Since $x_{e_a, v_a} = f(e)$ for all $v$ incident to $e$, the payoff of $v_a$ can be expressed as $\sum_{e \in E(v)} w(v_a, e_a) \cdot f(e)$. Consider the coalition $C$ formed by the edge agent $e_a$ and the vertex agents corresponding to the vertices in $e$. Consider the data exchange $\boldsymbol{y}$ obtained by setting $y_{e_a, v_a} = y_{v_a, e_a} = 1$ for all $v$ incident to $e$ in $G$. Also, recall that we are working under the assumption that all pairwise exchanges between the vertex agents and edge agents corresponding to the hyperedge $e$ have the same value in the exchange $\boldsymbol{x}$. By the construction of $u_{e_a}(\cdot)$, we have $u_{e_a}(\boldsymbol{y}) > u_{e_a}(\boldsymbol{x})$[2]. Since $\boldsymbol{x}$ is core-stable, at least one of the vertex agents, say $v_a$, must have $u_{v_a}(\boldsymbol{x}) \geq u_{v_a}(\boldsymbol{y})$. Therefore for any vertex agent $v_a$ in $C$, we have $u_{v_a}(\boldsymbol{y}) = w(e_a, v_a) \cdot 1$ (note

---

[1] We use the notation $a_+$ to represent $\max(a, 0)$.

[2] As $\boldsymbol{y}$ can be obtained by increasing all pairwise exchange by the same additive factor

that the cost is zero as the total data received is equal to 1). Since there exists an agent $v_a$ in $C$ with $u_{v_a}(\boldsymbol{x}) \geq u_{v_a}(\boldsymbol{y})$, we have,

$$w(e_a, v_a) = u_{v_a}(\boldsymbol{y}) \leq u_{v_a}(\boldsymbol{x}) = \sum_{e' \in E(v)} w(e'_a, v_a) f(e) \leq \hat{w} \sum_{e':e' \succeq e} f(e) + \tilde{w} \sum_{e':e' \prec e} f(e),$$

where $\tilde{w} = \max_{e' \in E(v): e' \prec e} w(e'_a, v_a)$ (equals zero if there is no $e' \in E(v)$ such that $e' \prec e$), and $\hat{w} = \max_{e' \in E(v)} w(e'_a, v_a)$. Observe that $\hat{w} \geq w(e_a, v_a) > \tilde{w}$. Now, substituting $\sum_{e':e' \prec e} f(e)$ as $1 - \sum_{e':e' \succeq e} f(e)$, we have

$$\hat{w} \cdot \sum_{e':e' \succeq e} f(e) + \tilde{w} \cdot \left(1 - \sum_{e':e' \succeq e} f(e)\right) \geq w(e_a, v_a) \Rightarrow \sum_{e':e' \succeq e} f(e) \geq \frac{w(e_a, v_a) - \tilde{w}}{\hat{w} - \tilde{w}}.$$

Given that $|E(v)| \leq 3$ for all $v$, we can set

$$w(e_a, v_a) = \begin{cases} d + H + 1 & e \text{ is } v\text{'s favorite edge} \\ d + H & e \text{ is } v\text{'s second favorite edge} \\ H & \text{otherwise,} \end{cases}$$

implying that $\sum_{e':e' \succeq e} f(e) \geq \frac{w(e_a, v_a) - \tilde{w}}{\hat{w} - \tilde{w}} \geq \frac{d}{d+1} \geq 1 - \epsilon$ for a sufficiently large $d$. This implies that $f$ is a stable matching. $\qquad\square$

## 4 The Pivoting Algorithm and Empirical Results

In this section, we adapt the pivoting algorithm [Sca67] to find a core-stable data exchange under sufficient conditions – when payoff functions are concave, and cost functions are convex, and coalitions are constrained to be of constant size. As we mentioned in Section 2.2, our game is balanced and therefore we can get a pivoting algorithm (PA) following the proof of existence of core for a balanced game theorem in Scarf's theorem [Sca67]. It is worth noting that, when the number of agents is not constant, even verifying whether a given data exchange is core-stable is coNP-complete, and we defer the proof to Appendix C.2.

### 4.1 The Pivoting Algorithm

We first define two matrices: *coalition matrix* $\mathbf{C}$ and the *utility matrix* $\mathbf{U}$, both with dimension $n \times m$. Given a coalition $S \subseteq [n]$, we aim to identify all possible utility vectors resulting from data exchanges within $S$. Since this set may be infinite, we instead consider a discretized approximation of the utility space. Suppose $\boldsymbol{v} \in V(S)$. We create the $k$th column in the coalition matrix $\mathbf{C}$: $C_{i,k} = 1$ if $i \in S$ and 0 otherwise, which is the characteristic vector of the coalition. Meanwhile, we insert a column into the utility matrix $\mathbf{U}$ at the same coordinate: $u_{i,k} = M$ for $i \notin S$ and $u_{i,k} = v_i$ for $i \in S$ where $M$ is a very large number[3]. Note that to compute the utility matrix, we need to find all achievable utility values in the grid. To this end, we assume that the payoff of every agent is uniformly bounded[4] by $B$. For a fixed $\epsilon > 0$, we create a grid of size $\epsilon$, so utility of every agent will be an integer multiple of $\epsilon$. Given a coalition of size $k$, there are at most $(B/\epsilon)^k$ possible utility vectors. Using the Ellipsoid method, we can approximate the feasibility of each in polynomial time, as shown in the following claim (proof deferred to Appendix C.1).

**Claim 5.** *Given a coalition $S \subseteq [n]$ and a utility profile $(v_i)_{i \in S}$, we can compute in polynomial time either a data exchange $\boldsymbol{x}$ such that $u_i(\boldsymbol{x}) \geq v_i - \epsilon$ for all $i \in S$ or return that no data exchange satisfies $u_i(\boldsymbol{x}) \geq v_i$ for all $i \in S$.*

After constructing the two matrices, we reduce finding core-stable data exchange to solving Scarf's Lemma (as elaborated in Appendix C.3). Then we apply PA. It maintains two evolving bases: a *cardinal basis* for $\mathbf{C}$ and an *ordinal basis* for $\mathbf{U}$, where each basis consists of a set of column indices. If the two bases differ, PA respectively performs the *cardinal pivot* step and the *ordinal pivot* step to adjust the two bases. These bases evolve iteratively until equal. As the total search space is finite, PA terminates in finite time. We defer a detailed description to Appendix C.4.

---

[3] The intuition behind choosing a large $M$ is that we do not want an agent outside the coalition block deviation to this coalition. In fact, we assign slightly different $M$ to these entries to make these entries non-identical.

[4] By scaling down the payoff functions, our hardness results still hold.

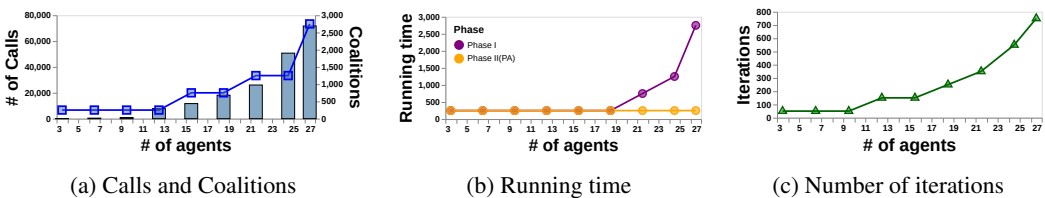

| (a) Calls and Coalitions | (b) Running time | (c) Number of iterations |

Figure 2: Performance of PA on the constructed data exchange instance.

## 4.2 Empirical Results

Next, we empirically test the performance of PA on real-world datasets. We construct a data exchange instance using the same road map dataset [roa] as the previous work [BGI+24b]. The graph contains 4.4K nodes and 9.6K edges. The data exchange instance is constructed as follows: each agent $i$ corresponds to a path $P_i$. The agents aim to estimate the delays on their roads. Assume each agent $i$ possesses $z_e^i$ random samples of the delay for every edge $e$ included in her path with identical variance $\sigma_e^2$. The payoff of agent $i$ is determined by the sum of the variance of the mean of the random samples of every edge that path $P_i$ includes. For every edge $e \in P_i$, the initial variance of the mean is given by $\sigma_e^2/z_e^i$. Hence, the initial sum of variance of agent $i$ is given by $\sum_{e \in P_i} \sigma_e^2/z_e^i$. In an exchange $\boldsymbol{x}$, $x_{i,j}$ represents agent $i$ will shares $x_{i,j} \cdot z_e^i$ data samples with agent $j$ for every $e \in P_i \cap P_j$. The payoff function $p_i(\cdot)$ of agent $i$ is induced by the decrease of the sum of variance, as follows:

$$p_i(\boldsymbol{x}) = \sum_{e \in P_i} \frac{\sigma_e^2}{z_e^i} - \sum_{e \in P_i} \frac{\sigma_e^2}{z_e^i + \sum_{j:e \in P_j, j \neq i} x_{j,i} \cdot z_e^j}.$$

Meanwhile, we assume that every agent $i$ incurs a cost of $c_i(\boldsymbol{x}) = \mu_i \sum_{e \in P_i} \mu_e \cdot z_e^i \sum_{j:e \in P_j, j \neq i} x_{i,j}$, where $\mu_e$ represents the cost of sharing per random sample of edge $e$

**Simulation Setup.** We run PA on the above data exchange instance to find a $0.1$-core-stable exchange, fixing each coalition size to 3, and varying the number of agents as $n = 3i$ for $i \in [9]$. We adopt the same method as [BGI+24b] to sample an agent $i$ from the road map: Sample a random node $u$ and then sample a length $t$ uniformly at random between 5 and the depth of the BFS tree rooted at $u$. Then we sample another node uniformly at random from all nodes within layer $t$, choose the shortest path from $u$ to $v$, and assign it to agent $i$. The variance of each edge, $\sigma_e$, is a random number in range $[0, 1]$, and $\mu_e$ is set as $(1 - \sigma_e) \cdot 10^{-3}$. In addition, every agent $i$ starts with $z_e^i$ random data samples for every edge in her path, where $z_e^i$ is chosen uniformly at random in the range $[4, 9]$. All experiments were run on a MacBook Pro with an Apple M3 Pro CPU and 18 GB RAM. The implementation uses Python 3.12 and SciPy [VGO+20] (v1.13.0) for concave optimization.

**Performance.** We evaluate PA's performance on the constructed data exchange instance using three metrics: (i) number of coalitions formed; (ii) number of concave optimization calls; and (iii) runtime for coalition matrix construction and pivoting. For each agent size, we repeat the construction and PA execution 20 times and report the mean. Results are shown in Figure 2, with the first subfigure displaying optimization calls and coalition sizes. Both reach the maximum when $n = 27$, where the number of calls and size of coalitions are respectively $7.17 \times 10^4$ and $2,914$. In addition, Figure 2b shows the time of constructing the coalition matrix (Phase I) and PA (Phase II). The most (least) expensive sample includes $9,198$ (resp. $533$) possible coalitions, and the time of constructing the coalition matrix is $9,821.54$ (resp. $151.36$) seconds, while the running time of the pivoting algorithm is $118.08$ (resp. $2.36$) seconds. The time of constructing the coalition matrix grows significantly in polynomial time, while the time of PA is much smaller and less affected by the number of agents. On average, PA terminates in $7.36$ seconds, ranging from $0.001$ seconds for $n = 3$ to $38.37$ seconds for $n = 27$. The third subfigure (Figure 2c) shows the number of iterations until termination. We observe a modest, seemingly quadratic growth in iterations with the number of agents. PA takes an average of $238.3$ iterations and up to $1434$ in the worst case.

# 5 Discussion and Conclusion

We introduced a general model of data exchange and studied the existence and computation of core-stable solutions within it. We identified interesting sufficient conditions for the existence of core-stable exchanges. We outline computational barriers and end with an algorithm that can be expected to be efficient in practice. We see this paper as an initiation for studying stability in data exchange economies. Currently, our model is very general in its assumptions, and we expect more efficient algorithms for some special cases, e.g., (i) accuracy functions from Gaussian inference or PAC learning (usually $p_i(\boldsymbol{x}_{-i} = 1 - 1/(\sum_j \tau_{ji} x_{ji}))$), and cost functions being linear, or (ii) when there are only constantly many payoff and cost functions, corresponding to fixed types of agents.

Another promising direction is to explore additional desiderata beyond stability. For instance, [BGI$^+$24a, ACGM25] investigate exchanges that satisfy both core-stability and a fairness criterion, where each agent's final payoff is proportional to their contribution to the payoffs of others. However, their model does not incorporate costs for data sharing– making core-stability alone trivial to achieve (set all $x_{ij} = 1$). An intriguing question is whether one can compute exchanges that are both core-stable and fair when agents incur costs for data-sharing. In essence, this involves integrating our cost-aware framework with the fairness-oriented approach of [BGI$^+$24a, ACGM25].

Finally, we believe that questions of stability merit investigation in more general models—particularly those involving *externalities*. Such externalities arise naturally in competitive environments, where collaboration between two agents may negatively impact the utility of others (e.g., collaborating agents can capture a greater share of the market). In these settings, we suspect that core-stability is unlikely to hold. However, it would be interesting to explore whether suitable mechanisms—such as structured rules for splitting the joint utility gains—can facilitate the existence of core-stable data exchanges despite the presence of externalities.

## Acknowledgements

The research of Bhaskar Ray Chaudhury and Jiaxin Song is supported by an NSF CAREER grant CCF No. 2441580. The research of Parnian Shahkar was supported by NSF grant CCF-2454115. The research of Pooja Kulkarni was supported by NSF grant CCF-2334461. Most of the work was done when the author, Pooja Ravi Kulkarni, was a student at UIUC. We also thank the anonymous reviewers for their helpful comments and suggestions.

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

# A Existence of Core-Stability

In this section, we provide a complete picture on the existence of core-stable data exchanges. In Appendix A.2, we show a core-stable data exchange always exists when (i) the payoff functions are concave and (ii) the cost functions are convex. From here on, we refer to the foregoing two conditions as *sufficient conditions*. Then we show that core-stable data exchange may not always exist if one of the two sufficient conditions is unsatisfied, even when we have only three agents.

## A.1 N-Person Game, Balanced Game

The definition of the core of an $n$-person game was proposed by Scarf [Sca67]. An $n$-person game with non-transferable utilities consists of $n$ agents (denoted by $N$) and a function $V(\cdot)$. Let $\mathbb{R}_{\geq 0}^S$ to be a subspace of $\mathbb{R}_{\geq 0}^n$ where the entries corresponding to coordinates indexed by $S$ can take values in $\mathbb{R}_{\geq 0}$, while the entries for coordinates not in $S$ are set to 0. For every subset $S$ of $N$, function $V(\cdot)$ returns a set of outcomes $V(S)$ consisting of a set of utility vectors, which belong to $\mathbb{R}_{\geq 0}^S$ and represent all the achievable utilities of agents in $S$ when they collaborate exclusively with other agents in $S$. Next, we introduce the notion of *balanced game*.

**Definition 4** (Balanced Game). A collection $T$ of subsets of agents is said to be *balanced*, if there exists an assignment $\{\delta_S\}_{S \in T}$ such that, for every agent $i \in N$, we have $\sum_{S:i \in S} \delta_S = 1$. We say a utility vector $\boldsymbol{u}$ is attainable by $S$ if $\boldsymbol{u} \in V(S)$. If $\boldsymbol{u}$ is a utility vector for $n$ agents, let $\boldsymbol{u}_S$ denote the projection of $\boldsymbol{u}$ onto the agents in $S$. A game *balanced* if and only if for any balanced collection $T$ and any $\boldsymbol{u}$, if $\boldsymbol{u}_S$ is attainable by all $S$ in $T$, then $\boldsymbol{u}$ is attainable by $N$.

The core of an $n$-person game is defined almost the same as core-stable exchange. A *core* is a utility vector that is attainable by the entire agent set $N$ and cannot be blocked by any coalition. Given a utility vector $\boldsymbol{u}$, if a coalition of agents can get a higher utility for all of its members, then the vector $\boldsymbol{u}$ is said to be *blocked* by that coalition. As shown in [Sca67], the core of any $n$-person *balanced* game exists if $V(\cdot)$ satisfies the following mild assumptions.

**Lemma 2** ([Sca67]). *The core of any balanced $n$-person game exists if the function $V(\cdot)$ is assumed that:*

- *For each $S \subseteq N$, $V(S)$ is a closed set;*

- *If $\boldsymbol{u} \in V(S)$ and $\boldsymbol{y} \in \mathbb{R}_{\geq 0}^S$ with $\boldsymbol{y} \leq \boldsymbol{u}$[5], then $\boldsymbol{y} \in V(S)$;*

- *The set of vectors in $V(S)$ in which each player in $S$ receives no less than the maximum that she can obtain by herself is nonempty and bounded.*

## A.2 Existence of core-stable data exchange under Sufficient Conditions

We first formalize the data exchange problem into the framework of an $n$-person game. The function $V(\cdot)$ is specified as follows: for every subset $S \subseteq N$, we define $V(S)$ as the set of nonnegative utility vectors that are attainable by the agent set $S$. Formally,

$$V(S) = \left\{ (u_i(\boldsymbol{x}))_{i \in N} : \boldsymbol{x} \in [0,1]^{S \times S} \text{ and } u_i(\boldsymbol{x}) \geq 0, \forall i \in N \right\},$$

where $[0,1]^{S \times S}$ denotes the set of all data exchanges among $S$, i.e., only shares between agents in $S$ can take values in $[0,1]$ while other fractions are forced to be zero. Next, we show the game is balanced when the payoff functions $p_i(\cdot)$ are concave and cost functions $c_i(\cdot)$ are convex, which would then imply that a core exists.

**Theorem 3.** *A core-stable data exchange always exists if the payoff functions $\{p_i\}_{i \in N}$ are concave, the cost functions $\{c_i\}_{i \in N}$ are convex.*

*Proof.* We demonstrate that the function $V(\cdot)$ meets the pre-conditions of Lemma 2. First, we claim that $V(S)$ is a closed set for any $S \subseteq N$. For any sequence $\{\boldsymbol{u}^k \in V(S)\}_{k=1}^{\infty}$ converging to some utility vector $\boldsymbol{u}^*$, we show $\boldsymbol{u}^* \in V(S)$. By the definition of $V(S)$, $\boldsymbol{u}^k$ is attainable by

---

[5]This is coordinate-wise comparison of the vectors. Given vectors $\boldsymbol{a}, \boldsymbol{b} \in \mathbb{R}^d$, $\boldsymbol{a} \leq \boldsymbol{b}$ if and only if $a_i \leq b_i$ for all $i \in [d]$.

---

**Algorithm 1:** Update the data exchange $\boldsymbol{x}$ iteratively

---

**1** **Input**: $\epsilon > 0$, a utility vector $\boldsymbol{y}$ and an exchange $\boldsymbol{x}$ on a specified agent set $S \subseteq N$;

**2** Let $k \leftarrow 0$ and $\boldsymbol{x}^0 \leftarrow \boldsymbol{x}$;

**3** **while** *there exists an agent $i \in S$ such that $u_i(\boldsymbol{x}^k) \geq y_i + \epsilon$* **do**

**4** $\quad$ Let $k \leftarrow k + 1$ and $\boldsymbol{x}^k$ be the copy of the last exchange;

**5** $\quad$ Pick an arbitrary agent $j \in S$ with $j \neq i$ and positive shares with agent $i$, i.e., $x_{j,i}^k > 0$;

**6** $\quad$ Decrease the share $x_{j,i}^k$ until $x_{j,i}^k = 0$ or $u_i(\boldsymbol{x}^k) = y_i$;

**7** **return** $\boldsymbol{x}^k$;

---

some exchange $\boldsymbol{x}^k$. As the space of data exchanges $[0,1]^{S \times S}$ is compact, there exists a subsequence $\{\boldsymbol{x}^{k(\ell)}\}_{\ell=1}^{\infty}$ converging to some exchange $\boldsymbol{x}^* \in V(S)$. By the continuity of the utility functions, we have $u_i(\boldsymbol{x}^*) = \lim_{\ell \to \infty} u_i(\boldsymbol{x}^{k(\ell)}) = u_i^*$ for any $i \in S$. Thus, $\boldsymbol{u}^*$ is attainable by agent set $S$.

Next, we show that if $\boldsymbol{u} \in V(S)$ and $\boldsymbol{y} \in \mathbb{R}_{\geq 0}^S$ with $\boldsymbol{y} \leq \boldsymbol{u}$, then $\boldsymbol{y} \in V(S)$. Suppose $\boldsymbol{u}$ is attainable by some exchange $\boldsymbol{x}$. We now run the procedure described in Algorithm 1, which iteratively adjusts the current exchange until the utility vector is quite close to $\boldsymbol{y}$. We then show that $\boldsymbol{y}$ is also included in $V(S)$ as follows. First, we show that the procedure always terminates.

**Claim 6.** *For any $\epsilon > 0$, Algorithm 1 will terminate in finite steps.*

*Proof.* Let $\eta_{j,i}(\boldsymbol{x})$ be the infinum of $t \in [0,1]$ such that $u_i(\boldsymbol{x} + t \cdot \boldsymbol{e}_{j,i}) \geq u_i(\boldsymbol{x}) + \epsilon$. If such $t$ does not exists ($u_i(\boldsymbol{x} + t \cdot \boldsymbol{e}_{j,i}) < u_i(\boldsymbol{x}) + \epsilon$ for any $t \in [0,1]$), we set $\eta_{j,i}(\boldsymbol{x}) = 1$. For every agent $i \in S$, we let $\eta_i(\boldsymbol{x}) = \min_{j \in S, j \neq i} \eta_{j,i}(\boldsymbol{x})$. We first prove the following property.

$$\inf_{\boldsymbol{x}} \eta_{j,i}(\boldsymbol{x}) > 0, \quad \text{for any } i \neq j \in S.$$

Suppose for contradiction that $\inf_{\boldsymbol{x}} \eta_{j,i}(\boldsymbol{x}) = 0$. Then, there exists a sequence of exchanges and values of $t$, $\{(\boldsymbol{x}^k, t_k)\}_{k=1}^{\infty}$, such that $u_i(\boldsymbol{x}^k + t_k \cdot \boldsymbol{e}_{j,i}) > u_i(\boldsymbol{x}^k) + \epsilon$ and $\lim_{k \to \infty} t_k = 0$. As the whole space of exchanges is compact, there exists a subsequence $\{\boldsymbol{x}^{k(\ell)}\}$ converging to some exchange $\boldsymbol{x}^*$ and $t_{k(\ell)} \to 0$. Also, $u_i(\boldsymbol{x}^{k(\ell)} + t_{k(\ell)} \cdot \boldsymbol{e}_{j,i}) > u_i(\boldsymbol{x}^{k(\ell)}) + \epsilon$ for any $\ell$. By taking the limit on both sides, we get the contradiction as $\epsilon > 0$ and $u_i$ is continuous. Therefore, we can further conclude that,

$$\inf_{\boldsymbol{x}} \eta_i(\boldsymbol{x}) = \inf_{\boldsymbol{x}} \min_{j \in S, j \neq i} \eta_{j,i}(\boldsymbol{x}) = \min_{j \in S, j \neq i} \inf_{\boldsymbol{x}} \eta_{j,i}(\boldsymbol{x}) > 0, \text{ for any } i \Rightarrow \min_{i \in S} \left( \inf_{\boldsymbol{x}} \eta_i(\boldsymbol{x}) \right) > 0.$$

Denote the minimum by $\Delta$. We can observe that all the shares $x_{j,i}$ are iteratively decreased at each round. If a step terminates until $u_i(\boldsymbol{x}^k) = y_i$, $x_{j,i}$ will be decreased by a value of at least $\Delta$ by the definition of $\eta_i(\boldsymbol{x})$. For that reason, we know that, at each round, one of the following two must happen: (1) one positive $x_{j,i}$ is turned to zero; (2) one $x_{j,i}$ decreases by at least $\Delta$. As the sum of $x_{j,i}$ is bounded by $|S|^2$, the number of iterations is at most $|S|^2 / \Delta + 1$, which is finite. $\qquad \square$

According to Claim 6, we can find a data exchange $\boldsymbol{x}(\epsilon)$ over $S$ for every $\epsilon > 0$, such that its utility vector $\boldsymbol{u} = (u_i(\boldsymbol{x}))_{i \in S}$ satisfies $y_i \leq u_i(x) \leq y_i + \epsilon$. As the set $V(S)$ is closed, by taking $\epsilon$ to $0^+$, we know that $\boldsymbol{y} \in V(S)$.

Thirdly, when no one in $S$ shares anything with another one, all the utilities will be zero. So $\boldsymbol{0}$ is attainable by $S$. Besides, as all the utility functions are continuous and the space of all exchanges is closed, the utility vectors of $V(S)$ are bounded. Thus, we can conclude that the function $V$ satisfies the conditions in Lemma 2.

Finally, we show that the data exchange game is balanced. For any balanced collection $T$ and any $\boldsymbol{u}$, if $\boldsymbol{u}_S$ is attainable by all $S$ in $T$, we need to show that $\boldsymbol{u}$ is also attainable by $N$. We let $x_{i,j}^S$ to be the share of agent $i$ to agent $j$ in the data exchange that attains utility vector $\boldsymbol{u}_S$. We then construct a new exchange $\boldsymbol{x}$ over $N$ by

$$x_{i,j} = \sum_{S \in T : i \in S} \delta_S \cdot x_{i,j}^S,$$

As $x_{i,j}^S \leq 1$ and $\sum_{S \in T: i \in S} \delta_S = 1$, $x_{i,j} \leq 1$, which satisfies the feasibility constraints. Besides, since $p_i$ is concave and $c_i$ is convex, the utility function $u_i(\boldsymbol{x})$ is concave, which means that

$$
\begin{aligned}
u_i(\boldsymbol{x}) = u_i\left( \left( \sum_{S \in T: i \in S} \delta_S \cdot x_{s,t}^S \right)_{s,t \in [n]} \right) & \\
\geq \sum_{S \in T: i \in S} \delta_S \cdot u_i(\boldsymbol{x}_{-i,i}^S) & \qquad \text{(By concavity of } p_i(\cdot) \text{ and convexity of } c_i(\cdot)) \\
= \sum_{S \in T: i \in S} \delta_S \cdot u_i = u_i\,. & \qquad \text{(As } \boldsymbol{u}_S \text{ is attainable by } S)
\end{aligned}
$$

Hence, $u_i(\boldsymbol{x}) \geq u_i$ for any $i \in N$. By the second property of $V(\cdot)$, $\boldsymbol{u}$ is also attainable by $N$. Thus, the data exchange game is balanced and has a core $\boldsymbol{u}$. By the definition of the core, we know $\boldsymbol{u}$ is attainable by some exchange $\boldsymbol{x}$ of the $n$ agents, and $\boldsymbol{u}$ is not blocked by any utility vector of $V(S)$ for any $S \subseteq [n]$. For any utility vector $\boldsymbol{u}'$ that is attainable by $S$ but not included in $V(S)$, we know there must exist an agent in $S$ receiving negative utility, which means that $S$ cannot form a deviating coalition either. Therefore, $\boldsymbol{x}$ is a core-stable data exchange. $\qquad \square$

### A.3 Non-existence of Core-Stable Data Exchange For Instances Not Satisfying Sufficient Conditions

We first provide more details of the construction discussed in Section 2.1. By the construction of the payoff function, one agent receives a positive payoff only when she receives a share larger than $1 - \epsilon$ from another agent. As the cost function is monotone, we can reduce a fraction $x_{i,j}$ to 0 if it is smaller than $1 - \epsilon$, which does not decrease the utility of any agent and hence, does not affect the core stability of the data exchange $\boldsymbol{x}$. Below, we assume that $x_{i,j}$ is either 0 or larger than $1 - \epsilon$. Observe that the denominator of the third term of the cost function, $\epsilon$, is assumed to be a sufficiently small constant, such that if $x_{i,\alpha}$ and $x_{i,\beta}$ are both larger than $1 - \epsilon$, the cost will be no less than $(1 - \epsilon)^2/\epsilon$, which is pretty larger than the maximum payoff agent $i$ can receive (i.e., $p_{i,\alpha} + p_{i,\beta}$). Hence, agent $i$ would have the incentive to deviate, which implies that every agent cannot have positive shares with both two other agents.

| Case | Exchange | Agents | $x_{1,2}$ | $x_{1,3}$ | $x_{2,1}$ | $x_{2,3}$ | $x_{3,1}$ | $x_{3,2}$ | $u_1$ | $u_2$ | $u_3$ |
|---|---|---|---|---|---|---|---|---|---|---|---|
| I | $\boldsymbol{x}$ | $\{1,2,3\}$ | 0 | 0 | 0 | 0 | 0 | 0 | 0 | 0 | 0 |
|  | $\boldsymbol{x}^U$ | $\{1,2\}$ | 1 | 0 | 1 | 0 | 0 | 0 | 0.25 | 0.125 | - |
| II | $\boldsymbol{x}$ | $\{1,2,3\}$ | $\geq 1-\epsilon$ | 0 | $\geq 1-\epsilon$ | 0 | 0 | 0 | $< 0.25+\epsilon$ | $< 0.125+\epsilon$ | 0 |
|  | $\boldsymbol{x}^U$ | $\{2,3\}$ | 0 | 0 | 0 | 1 | 0 | 1 | - | 0.25 | 0.125 |
| II | $\boldsymbol{x}$ | $\{1,2,3\}$ | 0 | $\geq 1-\epsilon$ | 0 | 0 | $\geq 1-\epsilon$ | 0 | $< 0.125+\epsilon$ | 0 | $< 0.25+\epsilon$ |
|  | $\boldsymbol{x}^U$ | $\{1,2\}$ | 1 | 0 | 1 | 0 | 0 | 1 | 0.25 | 0.125 | - |
| II | $\boldsymbol{x}$ | $\{1,2,3\}$ | 0 | 0 | 0 | $\geq 1-\epsilon$ | 0 | $\geq 1-\epsilon$ | 0 | $< 0.25+\epsilon$ | $< 0.125+\epsilon$ |
|  | $\boldsymbol{x}^U$ | $\{1,3\}$ | 1 | 0 | 1 | 0 | 0 | 1 | 0.125 | - | 0.25 |
| III | $\boldsymbol{x}$ | $\{1,2,3\}$ | $\geq 1-\epsilon$ | 0 | 0 | $\geq 1-\epsilon$ | $\geq 1-\epsilon$ | 0 | $< 0.5+\epsilon$ | $< 0.125+\epsilon$ | $< -0.125+\epsilon$ |
|  | $\boldsymbol{x}^U$ | $\{3\}$ | 0 | 0 | 0 | 0 | 0 | 0 | - | - | 0 |
| III | $\boldsymbol{x}$ | $\{1,2,3\}$ | 0 | $\geq 1-\epsilon$ | $\geq 1-\epsilon$ | 0 | 0 | $\geq 1-\epsilon$ | $< -0.125+\epsilon$ | $< 0.25+\epsilon$ | $< 0.5+\epsilon$ |
|  | $\boldsymbol{x}^U$ | $\{1\}$ | 0 | 0 | 0 | 0 | 0 | 0 | 0 | - | - |

Table 1: Deviation $(U, \boldsymbol{x}^U)$ for each case of data exchange $\boldsymbol{x}$. The row with $\boldsymbol{x}$ is the original exchange and the row with $\boldsymbol{x}^U$ is the deviation. Cells in gray are the utilities of agents in the coalition $U$.

Next, we construct a simple graph $G = (V, E)$ with three vertices, with each of them corresponding to an agent. There is an edge from $i$ to $j$ if agent $i$ shares more than $1 - \epsilon$ unit of data with agent $j$. Therefore, every agent has an out-degree of at most 1 and there is no source in the graph. It suffices to discuss the following three cases.

- **Case I:** Nobody shares anything, i.e., $x_{i,j} = 0$ for any $i, j \in [n]$.

- **Case II:** Only two agents are exchanging data. Thus, there are three possible subcases: (i) $E = \{(1,2), (2,1)\}$; (ii) $E = \{(1,3), (3,1)\}$, or; (iii) $E = \{(2,3), (3,2)\}$.

- **Case III:** All three agents are exchanging. By the properties of the graph, there are only two possible subcases: (i) $E = \{(1,2), (2,3), (3,1)\}$ or; (ii) $E = \{(1,3), (3,2), (2,1)\}$.

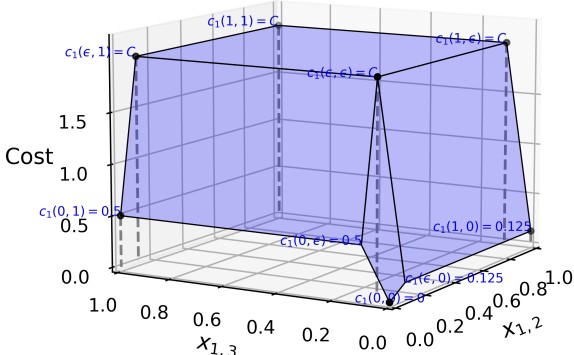

Figure 3: Cost function $c_1(\cdot)$ of agent 1

However, as shown in Table 1, for each case, there exists a coalition $U$ and an exchange $\boldsymbol{x}^U$ such that every agent in $U$ can improve her utility. In particular, for the first case, agent 1 and agent 2 have the incentive to form a coalition. For either of the second cases, there exists one agent of the two having the incentive to form a coalition with the other agent. For either of the third cases, there always exists one agent with a negative utility who can improve her utility by deviating alone. Therefore, we can conclude that it is impossible to find a core-stable exchange in both the three cases, which concludes the non-existence.

**Remark 1.** *It can be noticed that, in Table 1, each agent in the coalition $U$ improves her utility by at least $0.125 - \epsilon$ in each case, which even proves a stronger result – an $\alpha$-core-stable does not always exist in an arbitrary instance even when $\alpha$ is a positive constant number.*

Next, we show that when either of the two sufficient conditions – concavity of payoff functions or convexity of cost functions – is relaxed, core-stable data exchange may not always exist, even when having only three agents.

**Proposition 1.** *Even if both payoff and cost functions are continuous, core-stable data exchange does not always exist if either the concavity of payoff or the convexity of cost is relaxed.*

### A.3.1 Linear Payoff and Concave Cost

We have already provided an instance where core-stable data exchange does not exist when both the concavity of payoff and the convexity of cost functions are removed. Next, we show it also holds even when there are three agents and only the convexity of cost is removed.

Let the payoff functions of the three agents $p_1(\cdot), p_2(\cdot), p_3(\cdot)$ be defined as linear functions. In particular, $p_i(\boldsymbol{x}) = p_{i,\alpha} \cdot x_{i,\alpha} + p_{i,\beta} \cdot x_{i,\beta}$, where $\{\alpha, \beta\} = [3] \setminus \{i\}$ and $p_{i,j}$ is defined by Figure 1. The cost functions are defined as follows. Fix a sufficiently large constant $C > 0$. We take agent 1 as an example and start by defining her costs for some particular sharing $(x_{1,2}, x_{1,3})$:

$$c_1(x_{1,2}, x_{1,3}) = \begin{cases} 0 & \text{if } (x_{1,2}, x_{1,3}) = (0,0) \\ c_{1,3}/\epsilon \cdot x_{1,3} & \text{if } x_{1,2} = 0 \text{ and } x_{1,3} \leq \epsilon \\ c_{1,2}/\epsilon \cdot x_{1,2} & \text{if } x_{1,3} = 0 \text{ and } x_{1,2} \leq \epsilon \\ C & \text{if } x_{1,2} \geq \epsilon \text{ or } x_{1,3} \geq \epsilon \end{cases}$$

Then we define the costs for the remaining points $(x_{1,2}, x_{1,3})$ in $[0,1] \times [0,1]$. As shown in Figure 3, consider the four planes $P_1, P_2, P_3$ formed by connecting four points respectively, where plane $P_1$ is formed by points $(0,0,0)$, $(\epsilon, \epsilon, C)$, $(\epsilon, 0, c_{12})$, $(0, \epsilon, C)$, plane $P_2$ is formed by points $(\epsilon, 0, c_{1,2})$, $(1, 0, c_{1,2})$, $(1, \epsilon, C)$, $(\epsilon, \epsilon, C)$, plane $P_3$ is formed by points $(0, \epsilon, c_{1,3})$, $(0, 1, c_{1,3})$, $(\epsilon, 1, C)$, $(\epsilon, \epsilon, C)$, and plane $P_4$ is formed by points $(\epsilon, \epsilon, C)$, $(\epsilon, 1, C)$, $(1, \epsilon, C)$, $(1, 1, C)$.

Consider the union of the three planes $P = P_1 \cup P_2 \cup P3 \cup P_4$. Since their projections on the $x_{1,2} - x_{1,3}$ plane do not overlap (except the intersection line), for every $(x_{1,2}, x_{1,3}) \in [0,1] \times [0,1]$, there is just one possible $c$ such that $(x_{1,2}, x_{1,3}, c) \in P$. We then define $c_1(x_{1,2}, x_{1,3})$ as follows:

$$c_1(x_{1,2}, x_{1,3}) = c \text{ such that } (x_{1,2}, x_{1,3}, c) \in P.$$

Besides, as the constant $C$ is sufficiently large, we can observe that function $c_1$ is concave. We symmetrically define $c_2$ and $c_3$ as the above and they are concave for the same reason.

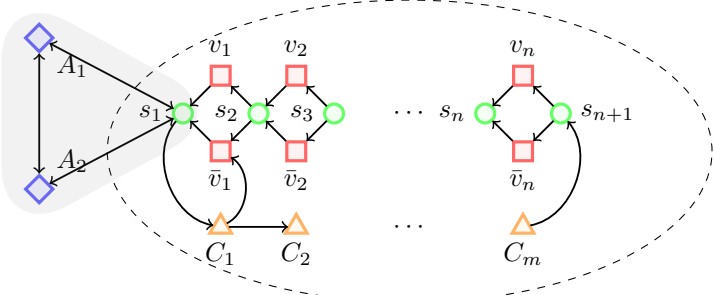

Figure 4: Illustration of reduction of Theorem 4, where each node corresponds to an agent.

Next, as $C$ is sufficiently large, every agent cannot share fractions of at most $\epsilon$ with the other two agents simultaneously.

Then we show a core-stable exchange still does not exist in this case. We prove it by contradiction. Assume $x$ is a core-stable exchange. Then we still discretize the data exchange $x$. If $x_{i,j} \leq \epsilon$, then we reduce it to zero, which only reduces the utility of agent $j$ by $\epsilon$. Otherwise, if $x_{i,j} > \epsilon$, then we increase it to 1, which does not affect agent $i$'s cost, since her two shares cannot be larger than $\epsilon$ simultaneously and the other share is reduced to zero. It can be noticed that $x_{i,j}$ turns to one or zero for any $i, j \in [3]$ after the discretization. In addition, the discretization operation only decreases the utility of each agent by $2\epsilon$.

However, since $2\epsilon$ is sufficiently small, our previous discussion demonstrates that there always exists a deviation $(U, x^U)$ that significantly improves the utility of every agent in $U$, no matter how the integral exchange performs, which means that there still exists a blocking $x$ before the discretization and contradicts assumption of core-stability of $x$.

### A.3.2 Convex Payoff and Convex Cost

We now construct a data exchange instance where core-stable exchange does not exist for convex payoff functions and convex cost functions. Set the number of agents as three and use the same payoff function as Section 2.1. Define the cost function as follows:

$$c_i(x) = \frac{1}{\epsilon} \cdot (x_{i,\alpha} + x_{i,\beta} - 1)_+ + c_{i,\alpha} \cdot x_{i,\alpha} + c_{i,\beta} \cdot x_{i,\beta},$$

where $\{\alpha, \beta\} = [3] \setminus \{i\}$. As the payoff functions are the same, we can still discretize the data exchange to make sure every $x_{i,j} = 0$ or $x_{i,j} > 1 - \epsilon$. It can be observed that, for every agent $i$, it is impossible that $x_{i,\alpha}$ and $x_{i,\beta}$ are both at least $1 - \epsilon$, since the incurred cost will become larger than $1/\epsilon \cdot (1 - 2\epsilon)$, which is much larger than the payoff she can receive. Using the same analysis before, we can conclude that there always exists a deviation from $x$.

## B   The Complexity of Finding Core-Stable Data Exchanges

### B.1   Hardness of Determining the Existence of a Core-Stable Data Exchange

**Theorem 4.** *It is* NP-*hard to determine the existence of a core-stable data exchange.*

**Overview of the Reduction.**   We reduce from the 3SAT problem, which is known to be NP-complete. Given a 3SAT instance, we create a data exchange instance with two parts, as illustrated in Figure 4, distinguished by shallow and dotted curves. The nodes represent the agents, and only agent $s_1$ appears in both areas. The lightly shaded area on the left corresponds to the instance discussed, where core-stable exchanges do not exist. In addition, we create a gadget using the input 3SAT instance in the right area. Each agent receives a positive payoff only if she simultaneously receives almost all data from every agent who points to her. In addition, we sets a threshold for every agent such that she incurs a large cost if the total fraction she shares exceeds the threshold.

Agent $s_1$ receives payoffs from both areas and takes the maximum as her final payoff and serves as a variable that determines whether a core-stable exchange exists. When the input 3SAT instance is a

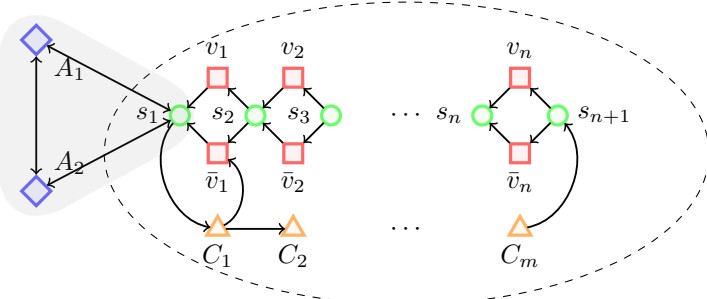

(a) Illustration of the directed graph gadget used in Theorem 4, where diamond nodes represent super agents, square nodes represent literal agents, circle nodes represent selection agents and triangle nodes represent clause agents. Clause $C_1$ is in form of $C_1 = (v_1 \vee \cdots)$. An arrow $i \to j$ represents data share $x_{i,j}$ affecting the payoff of agent $j$. The graph within the dotted ellipse represents the graph $G$.

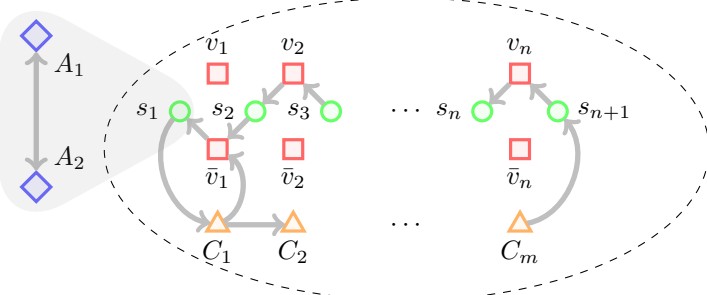

(b) The data exchange $\boldsymbol{x}^{\mathsf{Y}}$, where the truth assignment is $v_1 = \mathsf{false}, v_2 = \mathsf{true}, \ldots, v_n = \mathsf{true}$ and an arrow $i \to j$ in gray shadow represents agent $i$ sharing one unit of data with agent $j$.

**YES** instance, then we are able to construct a data exchange according to the assignment such that (almost) every agent in the right area receives the maximum payoff. Hence, no agent in the right area has an incentive to deviate from the data exchange. However, if the input 3SAT instance is a **NO** instance, the construction guarantees that no agent in the right area should exchange data with any other agent. Otherwise, the threshold requirement of some agent would be violated, which would lead to a large cost and give her an incentive to deviate. Therefore, the data exchange only happens in the left area, which is the instance we discussed in the last section, and we know that the core-stable exchange does not exist.

We now present the formal proof for Theorem 4.

*Proof.* We show a reduction from the 3SAT problem. Given a 3SAT instance with $n$ variables $v_1, \ldots, v_n$ and $m$ clauses $C_1, \ldots, C_m$, we construct a directed graph gadget $G = (V, E)$ (as shown in Fig. 5a) and then use it to construct a data exchange instance. For each variable $v_i$, we construct two literal vertices $v_i$ and $\bar{v}_i$ and a selection vertex $s_i$. For each clause $C_j$, we construct a clause vertex $C_j$. The vertices are connected as follows: (i) For each variable $v_i$, we create two arcs from literal vertex $v_i$ and $\bar{v}_i$ to $s_i$. Meanwhile, we create two arcs $(s_{i+1}, v_i)$ and $(s_{i+1}, \bar{v}_i)$; (ii) Then, we create a path from $s_1$ to $s_{n+1}$ that goes through all the clause vertices $C_1, \ldots, C_m$; (iii) Finally, for each clause $C_j$, if literal $v_i$ appears in $C_j$, we construct an arc from $C_j$ to $\bar{v}_i$. If $\bar{v}_i$ is in $C_j$, construct an arc from $C_j$ to $v_i$. For example, if $C_n$ is in form of $C_n = (v_1 \vee \cdots)$, then we construct a arc from $C_n$ to $\bar{v}_1$ in Fig. 5a. Observe that the out-degree of every clause vertex is exactly 4. We now construct a data exchange instance as follows: Let every vertex of $G$ correspond to a *normal agent*. For simplicity, we use the same notation to denote both the vertices and their corresponding agents. We slightly abuse $N$ to denote the set of normal agents. In addition, we introduce two *super agents* $A_1$ and $A_2$ and connect them with the selection agent $s_1$ with bi-directional arcs.

**Payoff Function.** The payoff function of each agent $a$ is in form of $p_a(\boldsymbol{x}) = \max\left(p_a^V(\boldsymbol{x}), p_a^A(\boldsymbol{x})\right)$, where $p_a^V(\boldsymbol{x})$ only depends on data shares from normal agents and $p_a^A(\boldsymbol{x})$ only depends on the shares

from the two super agents. In particular, $p_v^A(\boldsymbol{x}) = 0$ for any normal agent $v$ other than $s_1$. For any super agent $A \in \{A_1, A_2\}$, $p_a^V(\boldsymbol{x})$ is set to zero. We now proceed to define $p_a^V(\boldsymbol{x})$ and $p_a(\boldsymbol{x})$.

The payoff function $p_v^V(\boldsymbol{x})$ for every normal agent $v \in V$ is defined as follows:

- For each selection agent $s$, let $x_{v_1,s}$ and $x_{v_2,s}$ be the fractions of shares received from the agents with arcs incident to $s$. When $s = s_{n+1}$, we set $x_{v_1,s} = x_{v_2,s}$ as the share from $C_m$. Define $p_s^V(\boldsymbol{x}) = 1/\epsilon \cdot (\max(x_{v_1,s}, x_{v_2,s}) - (1 - \epsilon))_+$. Hence, the maximum payoff $p_s^V$ of a selection agent $s$ is 1.

- For any other agent $v$, let $p_v^V(\boldsymbol{x}) = \prod_{(u,v)\in E} 1/\epsilon \cdot (x_{u,v} - (1 - \epsilon))_+$. This implies that the maximum payoff these agents can receive is also 1.

Next, we define $p_a^A(\boldsymbol{x})$ for the two super agents $A_1, A_2$, and the selection agent $s_1$. Fix $\gamma > 0$ as a constant smaller than $\epsilon$. Consider the counter-example presented in Appendix A.3. We consider agents $A_1, A_2$, and $s_1$ to be the three agents 1, 2, and 3 (respectively) in that data exchange instance. Let $p_a^A(\boldsymbol{x})$ be the payoff function of any agent $a \in \{A_1, A_2, s_1\}$ from Appendix A.3, scaled by a factor of $\gamma$.

**Cost Function.** The cost function is also in the form of $c_a(\boldsymbol{x}) = c_a^V(\boldsymbol{x}) + c_a^A(\boldsymbol{x})$, where $c_a^V(\boldsymbol{x})$ and $c_a^A(\boldsymbol{x})$ are set as zero for any super agent and any normal agent (except for $s_1$).

The cost function $c_v^V(\boldsymbol{x})$ for every normal agent $v \in V$ consists of two parts: the first part is a sum of concave functions where each term becomes a small constant $\epsilon$ if $x_{v,u} > \epsilon$; while the second part is a convex function that becomes super large if the total out-shares is larger than a threshold. Formally, we set $c_v^V(\boldsymbol{x}) = \sum_{(v,u)\in E} \epsilon \cdot \min(1/\epsilon \cdot x_{v,u}, 1) + 1/\epsilon \cdot (\sum_{(v,u)\in E} x_{v,u} - \tau_v)_+$. Let $\tau_v = 3$ for any clause agent $v$ and $\tau_v = 1$ for any normal agent $v$. $c_a^A(\boldsymbol{x})$ is defined similarly. For any agent $a \in \{A_1, A_2, s_1\}$, her cost function $c_a^A(\boldsymbol{x})$ is inherited from Appendix A.3, scaled by a factor of $\gamma$.

When the input 3SAT instance is a **YES** instance, we define a data exchange $\boldsymbol{x}^\mathsf{Y}$ according to the assignment as follows: (i) The two super agents share one unit of data with each other; (ii) If a literal $\ell_i$ ($v_i$ or $\bar{v}_i$) is assigned true, let her share one unit of data with $s_i$ and let $s_{i+1}$ share one unit of data with her; (iii) Let selection agent $s_1$ share one unit of data with clause agent $C_n$ and every clause agent shares one unit of data with her adjacent agent; In particular, $C_n$ shares one unit of data with $C_{n-1}$, $C_{n-1}$ shares one unit of data with $C_{n-2}$, and so on; (iv) For each clause agent $C_j$, if a literal $\ell$ within it is assigned false, then let $C_j$ share one unit of data with the agent corresponding to the negation of $\ell$. We present a graphical explanation of $\boldsymbol{x}^\mathsf{Y}$ in Figure 5b.

**Claim 7.** *If the* 3SAT *instance is a* **YES** *instance, the data exchange $\boldsymbol{x}^\mathsf{Y}$ is a core-stable exchange.*

*Proof.* First, it can be verified that every agent has a positive utility in $\boldsymbol{x}^\mathsf{Y}$. Then we prove $\boldsymbol{x}^\mathsf{Y}$ is core-stable by contradiction. Suppose that a deviation $(U, \boldsymbol{x}^U)$ exists. We first discretize $\boldsymbol{x}^U$ as follows: for every $x_{i,j}$, if $x_{i,j} \leq 1 - \epsilon$, we can decrease $x_{i,j}$ to zero as it does not affect agent $j$'s utility. In addition, since all the cost functions are monotone, the operation weakly decreases the cost of agent $i$, which further increases agent $i$'s utility. Thus, the discretization operation does not affect the feasibility of the deviation.

After discretization, if $U$ contains some normal agent, we claim there must exist one selection agent $s$ still with positive out-shares in $\boldsymbol{x}^U$. Suppose for contradiction, all selection agents share nothing in $\boldsymbol{x}^U$. If $U$ contains clause agents, there must exist one clause agent in $U$ not receiving any shares in $\boldsymbol{x}^U$. Hence, the utility of that clause agent will be non-positive in $\boldsymbol{x}^U$, which gives her no incentive to deviate from $\boldsymbol{x}^\mathsf{Y}$. Otherwise, if $U$ does not contain any clause agent, any literal agent should not be included either, as the payoff of these agents only depends on the clause agents and selection agents. Loss of a clause agent makes a literal agent receive significantly less utility than in $\boldsymbol{x}^\mathsf{Y}$. These imply that $U$ should not contain any normal agent, and this contradicts our assumption. Therefore, there must exist one selection agent $s$ with positive shares in $U$. That means the cost of that selection agent is at least $\epsilon$ by the definition of her cost function. Meanwhile, as her maximum payoff is 1, her utility in $\boldsymbol{x}^U$ cannot exceed her payoff in $\boldsymbol{x}^\mathsf{Y}$, which causes a contradiction.

Last, if the coalition $U$ does not contain any normal agent, it can also be verified that the two super agents cannot strictly increase their utility simultaneously. □

When the input 3SAT instance is a **NO** instance, we prove the non-existence of core-stable exchange by contradiction. For the sake of contradiction, we assume a core-stable exchange $\boldsymbol{x}^{\mathsf{N}}$ exists. In fact, we find that, to guarantee core stability, the shares among the normal agents $V$ can only be zero, i.e., $x_{i,j}^{\mathsf{N}} = 0$ for any $i \neq j \in N$.

**Claim 8.** $x_{i,j}^{\mathsf{N}} = 0$ *for any two distinct normal agents* $i, j$ *if the* 3SAT *instance is a* **NO** *instance.*

*Proof.* Let $U$ be the set of normal agents who share positively with other normal agents. If $U$ is empty, the claim is proved. Assume that $U$ is nonempty. We next demonstrate that an agent $v$ in $U$ must exist with a zero payoff, which means she has negative utility since she has a positive share – which incurs a positive cost. Our discretization operation ensures that every agent in $U$ has an out-share of at least $1 - \epsilon$, which makes her incur a cost of at least $\epsilon$.

Since $\epsilon > \gamma$, it implies $p_v^A(\boldsymbol{x}) < \epsilon$ and $p_v^V(\boldsymbol{x})$ should be positive for any $v \in U$. By the definition of $p_v^V$, the fractional data shared on every edge incident to every clause agent or literal agent in $U$ should be larger than $1 - \epsilon$. Similarly, the fractional data shared on one of the incident edges of every selection agent in $U$ should also be larger than $1 - \epsilon$. Since $U$ is nonempty, by the structure of the directed graph gadget, the following agents should be included in $U$: (i) every selection agent; (ii) at least one of $v_i$ and $\bar{v}_i$ for any $i \in [n]$; (iii) every clause agent.

By the structure of the graph gadget, it can be observed that the following agents should be included in $U$: (i) every selection agent; (ii) at least one of $v_i$ and $\bar{v}_i$ for any $i \in [n]$; (iii) every clause agent. Next, we interpret $U$ as an assignment of the input 3SAT instance. If $v_i$ and $\bar{v}_i$ are both included, we only keep an arbitrary one. Then we create an assignment as follows: if $v_i$ is included, then $v_i$ is assigned true. Otherwise, if $\bar{v}_i$ is included, $v_i$ is assigned false. Since the 3SAT instance is a **NO** instance, no matter how the assignment is given, one clause $C_i$ will be unsatisfied, which means all the literals within it are assigned false. Hence, the negations of all three literals are included in $U$, and $C_i$ should share $1 - \epsilon$ unit of data with each of them. Meanwhile, $C_i$ should also share at least $1 - \epsilon$ unit of data with her adjacent clause agent (or selection agent), which makes the sum of fractions on her out-edges to be no less than $4(1 - \epsilon)$. Thus, her incurred cost will be at least $1/\epsilon \cdot (4 - 4\epsilon - 3) = 1/\epsilon \cdot (1 - 4\epsilon)$, which is much larger than her maximum payoff as $\epsilon$ is a sufficiently small constant. So there must exist one clause agent receiving negative utility in $U$, and such agent has no incentive to deviate, which leads to a contradiction. Thus, there must exist an agent $v$ in $U$ whose payoff is zero while her cost is positive, but such an agent has a negative utility and has no incentive to deviate, but this leads to contradiction as well. Therefore, $U$ must be empty, which concludes the claim. $\qquad\square$

Based on Claim 8, only data exchanges between $s_1$ and the two super agents $A_1$ and $A_2$ can be positive. However, by the analysis of the counter-example in Appendix A.3, a deviation always exists no matter how they exchange. Therefore, there is no core-stable data exchange when the 3SAT is a **NO** instance, which concludes the correctness of the reduction. $\qquad\square$

## B.2 PPAD-Hardness of Finding Core-Stable Exchange under the Sufficient Conditions

In this part, we show that the problem of finding a core-stable exchange is **PPAD**-hard for concave payoff functions and convex cost functions, where a core-stable exchange is shown to exist in Section 2.2. It is worth noting that, the hardness result still holds when the size of coalitions is restricted by a fixed constant.

We first introduce the problem of APPROXIMATE FRACTIONAL HYPERGRAPH MATCHING, where we are given a hypergraph $G = (V, E)$, a preference system $\mathcal{O} = (\succ_v)_{v \in V}$ and a number $\epsilon \in [0, 1]$. The goal is to determine whether there exists a fractional matching $f : E \to [0, 1]$ such that $\sum_{e: e \in E(v)} f(e) \leq 1$ for all $v \in V$. The matching is *stable* if there exists a vertex $v$ in $e$ such that $\sum_{e' \in E(v): e' \succeq_v e} f(e') \geq 1 - \epsilon$ for every edge $e$. A stable fractional matching is known to exist even if $\epsilon$ is set as zero [AF03]. Finding a stable fractional matching is **PPAD**-complete when $\epsilon$ is set as $1/2^{20|E|^4}$ [IK18, Theorem 4]. Furthermore, by applying their result to the setting in [Csá22], the **PPAD**-completeness still holds even when each edge has a size of exactly 3 and every vertex is incident to at most three edges.

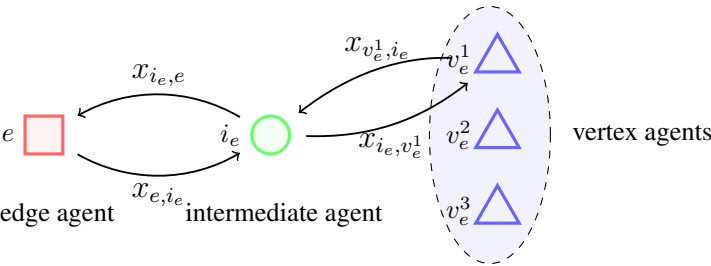

Figure 6: Graphical explanation of the edge gadget for the PPAD-hardness reduction.

### B.2.1 Overview of the Reduction

Given a hypergraph instance $G = (V, E)$ and a preference system $\mathcal{O}$, we construct the data exchange instance as follows. For every hyperedge $e$, we construct an *edge agent* (denoted by $e$ for simplification). Meanwhile, we create three *vertex agents* for the three vertices included in $e$, denoted by $v_e^1, v_e^2, v_e^3$. Note that if a vertex appears in multiple hyperedges, we do not create separate vertex agents for each occurrence. Instead, we create a unique vertex agent and simply refer to it by different names on different hyperedges. Additionally, we create one *intermediate agents* $i_e$ for every edge, who serve as the bridge facilitating data exchanges between edge agents and vertex agents. For every edge $e \in E$, we correspond intermediate agent $i_e^t$ to vertex agent $v_e^t$ for $t \in [3]$. We present a graphical illustration of the edge gadget in Fig. 6.

Fix $\epsilon > 0$ to be a sufficiently small constant such that $1 - 10\epsilon^{1/4} \geq 1 - 1/2^{20|E|^4}$ and $\epsilon < 10^{-3}$. Let $d = \epsilon^{-1/4}$, $H = \epsilon^{-1/2}$, and $\Gamma = 3(d + H + 1)/\epsilon$. Next, let us define the payoff and cost functions of the agents in the above data exchange instance.

**Payoff Functions.** For every vertex agent $v \in V$, we introduce a payoff function $p_v^e(\boldsymbol{x})$ for every hyperedge including it, and her total payoff is defined as the sum of these payoffs, i.e., $p_v(\boldsymbol{x}) = \sum_{e:v \in e} p_v^e(\boldsymbol{x})$. Consider a hyperedge $e \in E$ including $v$. Then we define $p_v^e(\boldsymbol{x})$ as follows:

$$
p_v^e(\boldsymbol{x}) = \begin{cases} (d + H + 1) \cdot x_{i_e, v} & \text{if } e \text{ is } v\text{'s favorite hyperedge,} \\ (d + H) \cdot x_{i_e, v} & \text{if } e \text{ is } v\text{'s second preferred hyperedge,} \\ H \cdot x_{i_e, v} & \text{if } e \text{ is } v\text{'s least preferred hyperedge .} \end{cases}
$$

The payoff function of every intermediate agent $i_e$ is $p_{i_e}(\boldsymbol{x}) = \min\left(x_{e, i_e}, x_{v_e^1, i_e}, x_{v_e^2, i_e}, x_{v_e^3, i_e}\right)$. In addition, the payoff function of the edge agent $e$ is defined as $p_e(\boldsymbol{x}) = x_{i_e, e}$.

**Cost Functions.** For every vertex agent $v$, define her cost function using the truncation function. The cost will be pretty large if her total shares with the intermediate agents exceed 1.

$$
c_v(\boldsymbol{x}) = \Gamma \cdot \left( \sum_{e:v \in e} x_{v, i_e} - 1 \right)_+ . \tag{3}
$$

Meanwhile, the cost function of an intermediate agent $i_e$ is defined as $c_{i_e}(\boldsymbol{x}) = (1 - \epsilon) \cdot \max\left(x_{i_e, e}, x_{i_e, v_e^1}, x_{i_e, v_e^2}, x_{i_e, v_e^3}\right)$. The cost function of edge agent $e$ is defined as $c_e(\boldsymbol{x}) = (1 - \epsilon) \cdot x_{e, i_e}$. We can check that all the payoff functions are concave and the cost functions are convex. Meanwhile, it can be observed that the utility of every agent is equal to zero when the data exchange $\boldsymbol{x}$ is $\boldsymbol{0}$.

### B.2.2 Mapping from Data Exchange to Fractional Hypergraph Matching

Next, we construct a hypergraph mapping from a core-stable data exchange $\boldsymbol{x}$. Since every agent receives a utility of zero in the data exchange $\boldsymbol{0}$, any core-stable data exchange must ensure that the utility of every agent is nonnegative. Otherwise, an agent receiving negative utility would have an incentive to deviate. We then adjust the fractions to ensure that no agent's utility decreases. Note that each agent's payoff and cost functions are monotonic with respect to every coordinate of the data exchange. For every intermediate agent $i_e$, since her cost function is defined as the maximum

of her out-shares $x_{i_e,e}, x_{i_e,v_e^1}, x_{i_e,v_e^2}, x_{i_e,v_e^3}$, we can adjust all the four fractions to be equal to the maximum of them, which weakly increase other agents' utility without increasing her cost. In addition, consider the edge agent $e$ and the three vertex agents corresponding to the intermediate agent. As the payoff of the intermediate agent is the minimum of the fractions of shares from the four agents, we can adjust the four fractions to be equal to the minimum of them, which weakly increases the utility of other agents without decreasing the payoff of the intermediate agent. As the utility of all agents does not decrease during the above adjustment, the tweaked data exchange $x$ is still core-stable. In the following proof, we assume that $x_{i_e,e} = x_{i_e,v_e^1} = x_{i_e,v_e^2} = x_{i_e,v_e^3}$ and $x_{e,i_e} = x_{v_e^1,i_e} = x_{v_e^2,i_e} = x_{v_e^3,i_e}$ for every intermediate agent $i_e$. Denote the former value by $f^-(e)$ and the latter value by $f^+(e)$. We first have the following lemma of the relationship between the two fractions $f^-(e)$ and $f^+(e)$ for every hyperedge $e$.

**Lemma 3.** *For every hyperedge $e \in E$, we have $(1 - \epsilon)f^+(e) \le f^-(e) \le f^+(e)/(1 - \epsilon)$.*

*Proof.* First observe that, the utility of the edge agent $e$ is given by

$$p_e(\boldsymbol{x}) - c_e(\boldsymbol{x}) = f^-(e) - (1 - \epsilon) \cdot f^+(e),$$

which should be nonnegative as the data exchange is core-stable. This implies that $f^-(e) \ge (1 - \epsilon) \cdot f^+(e)$. In addition, the utility of the intermediate agent $i_e$ is given by

$$p_{i_e}(\boldsymbol{x}) - c_{i_e}(\boldsymbol{x}) = f^+(e) - (1 - \epsilon) \cdot f^-(e),$$

which is also nonnegative. This implies that $f^+(e) \ge (1 - \epsilon) \cdot f^-(e)$ and concludes the lemma. $\square$

**Lemma 4.** *For every vertex $v \in V$, we have $\sum_{e:v \in e} f^+(e) \le (1 + \epsilon)$.*

*Proof.* Observe that, the utility of every vertex agent $v$ is given by

$$p_v(\boldsymbol{x}) - c_v(\boldsymbol{x}) = \sum_{e:v \in e} p_v^e(\boldsymbol{x}) - c_v(\boldsymbol{x}) \le \sum_{e:v \in e} (d + H + 1) - \Gamma \cdot \left( \sum_{e:v \in e} f^+(e) - 1 \right)_+$$

$$\le 3 \cdot (d + H + 1) - \Gamma \cdot \left( \sum_{e:v \in e} f^+(e) - 1 \right)_+, \quad \text{(the degree of $v$ is at most 3)}$$

which should be nonnegative. As $\Gamma = 3(d + H + 1)/\epsilon$, then we have $\sum_{e:v \in e} f^+(e) \le 1 + \epsilon$. $\square$

Now we are ready to construct the fractional hypergraph matching on that hypergraph. Define the flow $f : E \to [0, 1]$ as follows: for every edge $e \in E$, we let $f(e) = \max(f^+(e), f^-(e)) \cdot \frac{1-\epsilon}{1+\epsilon}$, which is valid fractional matching by Lemma 4.

### B.2.3   From Core-Stability to Stable Matching

Finally, we show the fractional matching constructed above is a stable fractional matching. For every hyperedge $e \in E$, if $f(e) \ge \frac{1-\epsilon}{1+\epsilon}$, then the stability constraint is met by every vertex $v$ included in the hyperedge since $\sum_{e':e' \succeq e} f(e') \ge f(e)$ and $\epsilon$ is assumed to be $2\epsilon < 2^{20|V|^4}$. Below we consider the case when $f(e) < \frac{1-\epsilon}{1+\epsilon}$. Then we can see that

$$\max(f^+(e), f^-(e)) \le f(e) \cdot \frac{1+\epsilon}{1-\epsilon} < 1$$

Denote the maximum of $f^+(e)$ and $f^-(e)$ by $x_e^*$. Let us define $\Delta_e = 1 - x_e^*$. This variable is positive since $\max(f^+(e), f^-(e)) < 1$. Consider a coalition $U$ consisting of the edge agent $e$, the intermediate agents $i_e$ and the three vertex agents $v_e^1, v_e^2, v_e^3$. Define the data exchange $\boldsymbol{x}^U$ as $x_{s,t}^U = x_{s,t} + \Delta_e$ for every $s, t \in U$. Observe that the edge agent and the intermediate agent all receive higher utility in deviation.

**Claim 9.** $u_e(\boldsymbol{x}^U) > u_e(\boldsymbol{x})$ and $u_{i_e}(\boldsymbol{x}^U) > u_{i_e}(\boldsymbol{x})$.

*Proof.* For the intermediate agent $i_e$, as all the in-shares (out-shares) are shifted by $\Delta_e$ simultaneously. Her payoff increases by $4\Delta_e$ while her cost increases by $4(1 - \epsilon)\Delta_e$. Thus, her total payoff increases by $4\epsilon\Delta_e$. Similarly, the payoff of the edge agent $e$ increases by $\Delta_e$ while her cost increases by $(1 - \epsilon)\Delta_e$. Therefore, her total payoff increases by $\epsilon\Delta_e$. $\square$

Since $x$ is a core-stable exchange, there must exist an agent in $U$ receiving less utility in $x^U$. By Claim 9, such agent must be one of the three vertex agents. Denote the agent by $v$ for simplicity. Let $f(e_1), f(e_2)$ and $f(e_3)$ respectively be the flows for $e_1, e_2, e_3$ respectively being vertex $v$'s most favorite, second favorite, and least favorite edge. Hence, we know $f(e_1) + f(e_2) + f(e_3) \leq 1$. We now discuss the following three cases according to the ranking of $e$ in vertex $v$'s preference order $\succ_v$.

**Case I: $e$ is $v$'s favorite edge.** In the new data exchange $x^U$, we know the maximum fraction is equal to 1 as $\Delta_e = 1 - x_e^*$. Hence, it holds that $x_{i_e,v}^U = x_{i_e,v} + \Delta_e \geq (1 - \epsilon)x_e^* + \Delta_e \geq (1 - \epsilon) \cdot (x_e^* + \Delta_e) = 1 - \epsilon$. Meanwhile, since the total out-shares of agent $v$ is $x_{v,i_e}$, which is at most 1 and does not exceed the threshold of Equation (3), her incurred cost is zero. Thus, the utility of agent $v$ in $x^U$, $u_v(x^U)$ is at least

$$u_v(x^U) \geq (1 - \epsilon) \cdot (d + H + 1).$$

On the other hand, we can observe that the utility of agent $v$ in exchange $x$ is at most

$$
\begin{aligned}
u_v(x) &= p_{e^1}(x) + p_{e^2}(x) + p_{e^3}(x) \\
&= f^-(e_1) \cdot (d + H + 1) + f^-(e_2) \cdot (d + H) + f^-(e_3) \cdot H \\
&\leq \frac{1 + \epsilon}{1 - \epsilon} \cdot (f(e_1) \cdot (d + H + 1) + f(e_2) \cdot (d + H) + f(e_3) \cdot H) \\
&\leq \frac{1 + \epsilon}{1 - \epsilon} \cdot (f(e_1) \cdot (d + H + 1) + (1 - f(e_1)) \cdot (d + H))) \\
&= \frac{1 + \epsilon}{1 - \epsilon} \cdot (f(e_1) + d + H) .
\end{aligned}
$$

Since $u_v(x) \geq u_v(x^U)$, it follows that

$$f(e_1) \geq \frac{(1 - \epsilon)^2}{1 + \epsilon} - (d + H) \cdot \left(1 - \frac{(1 - \epsilon)^2}{1 + \epsilon}\right) \geq 1 - 4\epsilon - 3\epsilon \cdot (d + H) \geq 1 - 10\epsilon^{1/2} .$$

**Case II: $e$ is $v$'s second favorite edge.** In this case, the utility of agent $v$ in data exchange $x^U$ is at least $u_v(x^U) \geq (d+H) \cdot (1-\epsilon)$. Since $f(e_1)+f(e_2)+f(e_3) \leq 1$, we have $f(e_3) \leq 1-f(e_1)-f(e_2)$. Thus, the utility of agent $v$ in $x$ is at most

$$
\begin{aligned}
u_v(x) &\leq f^-(e_1) \cdot (d + H + 1) + f^-(e_2) \cdot (d + H) + f^-(e_3) \cdot H \\
&\leq \frac{1 + \epsilon}{1 - \epsilon} \cdot (f(e_1) \cdot (d + H + 1) + f(e_2) \cdot (d + H) + f(e_3) \cdot H) \\
&\leq \frac{1 + \epsilon}{1 - \epsilon} \cdot (f(e_1) \cdot (d + H + 1) + f(e_2) \cdot (d + H) + (1 - f(e_1) - f(e_2)) \cdot H) \\
&\leq \frac{1 + \epsilon}{1 - \epsilon} \cdot ((f(e_1) + f(e_2)) \cdot (d + 1) + H)
\end{aligned}
$$

For this reason, we further derive that

$$
\begin{aligned}
f(e_1) + f(e_2) &\geq \frac{d}{d + 1} \cdot \frac{(1 - \epsilon)^2}{1 + \epsilon} - \left(1 - \frac{(1 - \epsilon)^2}{1 + \epsilon}\right) \cdot H \\
&\geq (1 - \epsilon^{1/4}) \cdot (1 - 3\epsilon) - 3\epsilon \cdot H \geq 1 - 7\epsilon^{1/4} .
\end{aligned}
$$

**Case III: $e$ is $v$'s least favorite edge.** In this case, we know the utility of agent $v$ in $x^U$ is at least $u_{v_e}(x^U) \geq H \cdot (1 - \epsilon)$. Meanwhile, the utility of agent $v_e$ in $x$ is at most

$$
\begin{aligned}
u_v(x) &\leq (f^-(e_1) + f^-(e_2) + f^-(e_3)) \cdot (d + H + 1) \\
&\leq \frac{1 + \epsilon}{1 - \epsilon} \cdot (f(e_1) + f(e_2) + f(e_3)) \cdot (d + H + 1),
\end{aligned}
$$

which further implies that

$$f(e_1) + f(e_2) + f(e_3) \geq \frac{(1 - \epsilon)^2}{1 + \epsilon} \cdot \frac{H}{d + H + 1} \geq (1 - 3\epsilon) \cdot (1 - \epsilon^{1/4}) \geq 1 - 4\epsilon^{1/4} .$$

We can observe that, in all three cases, the total flow on edges where $v$ has weakly better preference than $e$ is at least $1 - 10\epsilon^{1/4}$, which is larger than $1 - 1/2^{20|V|^4}$ by the setting of $\epsilon$, which indicates that the hypergraph matching is stable.

## C  Pivoting Algorithm

### C.1  Complexity of Construction of Coalition Matrix

**Claim 5.** *Given a coalition $S \subseteq [n]$ and a utility profile $(v_i)_{i \in S}$, we can compute in polynomial time either a data exchange $\boldsymbol{x}$ such that $u_i(\boldsymbol{x}) \geq v_i - \epsilon$ for all $i \in S$ or return that no data exchange satisfies $u_i(\boldsymbol{x}) \geq v_i$ for all $i \in S$.*

*Proof Sketch.* For each agent $i \in S$, we create a new utility function $u_i'(\cdot)$ that caps the original utility function at $v_i$: $u_i'(\boldsymbol{x}) = \min\{u_i(\boldsymbol{x}), v_i\}$. Next, consider the following concave program.

$$\max \quad \sum_{i \in S} u_i'(\boldsymbol{x})$$
$$\text{subject to} \quad 0 \leq x_{i,j} \leq 1, \quad \forall i, j \in S$$

Let $\mathcal{K} = \{\boldsymbol{x} : x_{i,j} \in [0,1], \forall i, j \in S\}$ be the feasible domain. We then apply the Ellipsoid method to find an approximate solution with a distance of at most $\epsilon$ to the optimal solution. By [Vis21, Theorem 13.1], it suffices to find a first-order oracle for $\sum_{i \in S} u_i'(\boldsymbol{x})$. As $u_i'(\cdot)$ is the minimum of $u_i(\cdot)$ and $v_i$, its supergradient can be answered in $O(1)$ time using [Sho12, Theorem 1.13].

Denote the solution found by the Ellipsoid method by $\boldsymbol{x}$ and the optimal solution by $\boldsymbol{x}^*$. Then

$$\sum_{i \in S} u'(\boldsymbol{x}) \geq \sum_{i \in S} u'(\boldsymbol{x}^*) - \epsilon.$$

We output $\boldsymbol{x}$ if $\sum_{i \in S} u_i'(\boldsymbol{x}) \geq \sum_{i \in S} v_i - \epsilon$ and **NO** otherwise. Below, we demonstrate the desirable property of the output answer. If a data exchange $\boldsymbol{x}$ is returned, then $\sum_{i \in S} u_i'(\boldsymbol{x}) \geq \sum_{i \in S} v_i - \epsilon$. Since $u_i'(\boldsymbol{x}) \leq v_i$ holds for every $i \in S$, then $u_i'(\boldsymbol{x}) \geq v_i - \epsilon$, which means $u_i(\boldsymbol{x}) \geq v_i - \epsilon$. If the output is **NO**, we prove by contradiction that no data exchange can meet $u_i(\boldsymbol{x}) \geq v_i$ for every agent $i$. If such data exchange $\boldsymbol{x}$ exists, then the optimal solution is at least $\sum_{i \in S} v_i$. Hence, as the approximation ratio is set as $\epsilon$, the found solution will have an objective value of at least $\sum_{i \in S} v_i - \epsilon$, which violates the assumption that the output answer is **NO**. Therefore, the claim is proved. $\square$

### C.2  Hardness of Verifying Core-Stability

**Theorem 10.** *It is* coNP*-complete to verify whether an exchange $\boldsymbol{x}$ is core-stable when the payoff functions are linear and the cost functions are convex.*

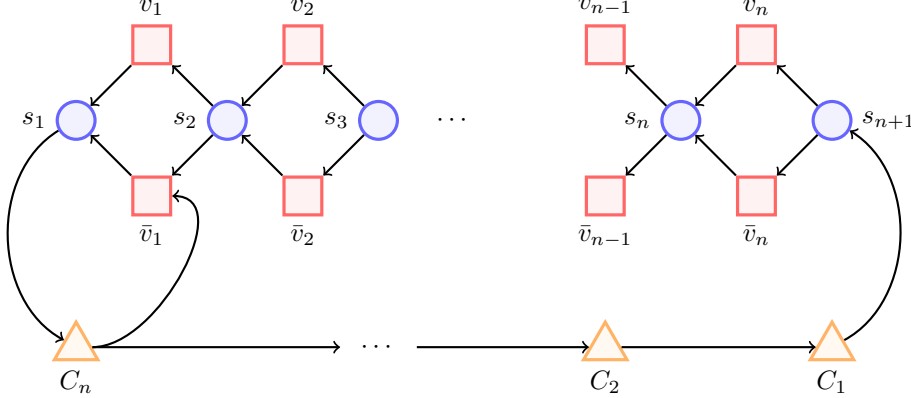

Figure 7: Construction of the graph gadget for the reduction from 3SAT to VERIFY CORE-STABILTY. The blue round nodes represent the selection vertices, the red square nodes represent the variable vertices, and the orange triangle nodes represent the clause vertices. Each clause agent has an outgoing edge to the negation of each literal it contains. For example, if $C_n = (v_1 \vee \cdots)$, there will be a arc $(C_n, \bar{v}_1)$ constructed.

*Proof.* The problem VERIFY CORE-STABILTY is in coNP since once we are given another data exchange $\boldsymbol{x}'$ and a coalition that can deviate to, it can be verified in polynomial time whether it blocks $\boldsymbol{x}$. Next, we show the coNP-hardness by reducing from the 3SAT problem. Given a 3SAT problem instance with $m$ clauses $\mathcal{C} = (C_i)_{i \in [m]}$ and $n$ variables $\mathcal{V} = (v_i)_{i \in [n]}$, we construct a data exchange instance $\mathcal{D}$ as follows.

First, we construct a graph gadget and use it to construct a data exchange instance then. For each variable $v_i$, we construct two literal vertices $v_i$ and $\bar{v}_i$ and a selection vertex $s_i$. For each clause $C_j$, we construct a vertex $C_j$. Then we connect these vertices as follows (see Figure 7): For each variable $v_i$, we create two arcs from literal vertex $v_i$ and literal vertex $\bar{v}_i$ to $s_i$. Meanwhile, we create arcs $(s_{i+1}, v_i)$ and $(s_{i+1}, \bar{v}_i)$. Then, we create a path from $s_1$ to $s_{n+1}$ that goes through all the clause vertices $C_n, \ldots, C_1$. Moreover, for each clause $C_j$, if variable $v_i$ occurs positively within $C_j$, then we construct an arc from $C_j$ to $\bar{v}_i$. Else, if $v_i$ occurs negatively within $C_j$, we construct an arc from $C_j$ to $v_i$. For example, in Figure 7, if $C_n$ is in form of $C_n = (v_1 \vee \cdots)$, then we construct an arc from $C_n$ to $\bar{v}_1$. Hence, we observe that, for each clause vertex, its out-degree is exactly 4 and its in-degree is 1.

Now, we construct the data exchange instance $\mathcal{D}$. Denote the set of all vertices by $V$ and the out-degree (in-degree) of each vertex $v$ by $d_v^{out}$ ($d_v^{in}$). The data exchange instance involves two types of agents: (1) The first is one *generous agent*, denoted by $g$; (2) The second is a set of agents $\{a_v\}_{v \in V}$, where $V$ is the set of all vertices in the graph gadget. We call these "normal" agents. In what follows, when referring to the agents, we will alternatively use the label of the corresponding graph vertex.

The payoff functions and cost functions are constructed as follows: The generous agent $g$ receives no payoff from the normal agents and incurs no cost when sharing data. Her utility is always zero in any data exchange. For each normal agent, we define her payoff function as follows.

**Payoff Functions.** Fix a small constant $\epsilon$ such that $0 < \epsilon < 1/4$. Define the function $p_i(\boldsymbol{x})$ for each normal agent as the sum of payoffs from the following two sources:

- *Payoff from the generous agent*: For every selection vertex $i = s_j$ for $j \in [n]$ and every clause vertex $i = C_j$ for $j \in [n]$, we define $p_i^g(\boldsymbol{x}) = (1 - \epsilon) \cdot x_{g,i}$. For every literal variable $\ell$, we let $p_\ell^g(\boldsymbol{x}) = (d_\ell^{in} - \epsilon) \cdot x_{g,\ell}$.

- *Payoff from other normal agents*: As mentioned, the generous agent gets no payoff from the shares of these normal agents. For each normal agent $i \in V$, we define $a_i^V(\boldsymbol{x}) = \sum_{(j,i) \in E} x_{j,i}$. That means, for each normal agent $p_i$, her payoff from $V$ is equal to the sum of the fraction of shares from agents that are adjacent to her in the graph gadget.

**Cost Functions.** Next, we define the cost functions for each agent. The cost function of the generous agent $g$ is defined as the constant zero. The cost functions of the normal agents are given as follows:

$$\triangleright \text{ Selection/Literal agent} \qquad c_i(\boldsymbol{x}) = \frac{1}{\mu} \left( \sum_{j \in V} x_{i,j} - 1 \right)_+$$

$$\triangleright \text{ Clause agent} \qquad c_i(\boldsymbol{x}) = \frac{1}{\mu} \left( \sum_{j \in V} x_{i,j} - 3 \right)_+$$

where $\mu$ is a sufficiently small constant smaller than $1 - 4\epsilon$. The idea behind the construction is that each normal agent will incur a large cost if the total fraction of her shares with other agents is larger than a threshold.

We construct the two oracles $\mathcal{U}$ and $\nabla \mathcal{U}$ as follows: We answer $\mathcal{U}(i, \boldsymbol{x})$ as the utility $p_i(\boldsymbol{x}) - c_i(\boldsymbol{x})$ as constructed above, which takes constant time since the out-degree of every vertex is bounded. Additionally, as every utility function can be rewritten as the maximum of $O(1)$ linear functions, we can figure out the supergradient $\nabla u_i(\boldsymbol{x})$ in $O(1)$ time.

Now we show that deciding whether the following data exchange $\boldsymbol{x}^0$ is $\alpha$-core-stable is equivalent to deciding whether the input 3SAT instance is satisfiable, where

$$\boldsymbol{x}^0 = (x_{i,j}) \quad \text{where } x_{i,j} = \begin{cases} 1 & \text{if } i = g, \\ 0 & \text{otherwise.} \end{cases}$$

Note that the generous agent is sharing her entire data with every normal agent, while the fractions of all other shares are set to zero. Hence, in $\boldsymbol{x}^0$, the utility of the generous agent is equal to zero, i.e., $u_g(\boldsymbol{x}^0) = 0$. In addition, every normal agent only gets data from the generous agent. Each selection agent or each clause agent receives a utility of $1 - \epsilon$ while each literal agent $\ell$ receives $u_\ell(\boldsymbol{x}^0) = d_\ell^{in} - \epsilon$.

Next, we show a **YES** to **NO** mapping and a **NO** to **YES** mapping. Let us first show the former one. As the input 3SAT instance is a **YES** instance, there exists a truth assignment such that each clause is satisfied. We then define a deviation $(U, \boldsymbol{x}^U)$ containing all the literal agents assigned true, all selection agents, and all clause agents as follows. If a variable $v_i$ is assigned as true, then we let $x_{v_i, s_i} = 1$ and $x_{\bar{v}_i, s_i} = 0$. Meanwhile, we let $s_{i+1}$ share 1 to $v_i$ and 0 to $\bar{v}_i$ for every $i \in [n]$. If a variable $v_i$ is assigned as false, then we define the shares symmetrically. Let selection agent $s_1$ share one unit of data with clause agent $C_n$, and every clause agent shares one unit of data with her adjacent agent. In particular, $C_n$ shares one unit of data with $C_{n-1}$, $C_{n-1}$ shares one unit of data with $C_{n-2}$, and so on. In addition, for each clause agent, $C_j$, for each literal $\ell$ it contains, if the literal is assigned as false, then we let $C_j$ share 1 unit of data with the agent corresponding to the negation of $\ell$. For literal agents not assigned true in the assignment, we just throw them away and only include the other agents in $U$.

**Proposition 2.** *The deviation* $(U, \boldsymbol{x}^U)$ *blocks the original exchange* $\boldsymbol{x}^0$.

*Proof.* As the input 3SAT instance is a **YES** instance, we know that each clause is satisfied. Hence, for each clause agent, at least one literal of it is assigned true. So, the total fractions of shares of the clause agent are at most $(3 - 1) + 1 = 3$, which is no more than the threshold of the cost function. Hence, her cost will still be equal to zero. In addition, as she also receives one unit of data from the adjacent agent, her utility will be equal to 1.

For each literal agent $\ell = v_j$ or $\bar{v}_j$ included in $U$, we know $s_{j+1}$ shares 1 unit of data to her, and any clause agent containing its negation also shares one unit of data to her. Thus, her total utility will be equal to $d_\ell^{in}$. Also, as she just shares one unit of data with the selection agent, it does not exceed the threshold of the cost function. Thus, her utility is also more than her old utility $d_\ell^{in} - \epsilon$.

Finally, we know each selection agent also receives an entire dataset from one of the adjacent literal agents and shares one unit of data with a literal agent. Hence, her utility is equal to $1 - 0 = 1$, which is also larger than her old utility in $\boldsymbol{x}^0$ (which is $1 - \epsilon$). $\qquad\square$

Now, we prove the remaining part of the reduction – the **NO** to **YES** mapping. If the input 3SAT instance is a **NO** instance, we now prove the data exchange is core-stable by contradiction. Suppose there exists a possible deviation $(U, \boldsymbol{x}^U)$ from $\boldsymbol{x}^0$. First, as the generous agent always has the utility of zero, she does not have the incentive to deviate. Hence, $U$ must be a subset of $V$.

Next, we demonstrate that when $U$ is a subset of $V$, it cannot guarantee that every agent in $U$ receives a higher utility than in $\boldsymbol{x}^0$. If $U$ includes one selection agent (denoted by $s_j$ without loss of generality), as she has the incentive to deviate, she should get more than $1 - \epsilon$ in the data exchange $\boldsymbol{x}^U$. Hence, one of $v_j$ and $\bar{v}_j$ must be included in $U$ as $\epsilon > 0$. Similarly, $s_{j+1}$ should also be included in $U$, and continuing this process, we can eventually conclude that all selection agents and clause agents should be included in $U$. In addition, at least one of the two literal agents $v_i$ and $\bar{v}_i$ is included for every $i \in [n]$. We arbitrarily choose one from each pair, and next construct an assignment $\varphi$ for the input 3SAT instance as follows. If $v_i$ is included, then we assign the variable $v_i$ as false, $\varphi(v_i) = $ false. Otherwise, we assign it as true, $\varphi(v_i) = $ true. For every included literal agent, her utility in the old data exchange $\boldsymbol{x}^0$ is $d^{in} - \epsilon$. Hence, as she has the incentive to deviate, she has received at least a $1 - \epsilon$ fraction from every clause agent adjacent to her. As the input 3SAT instance is a **NO** instance, there must exist a clause $C_i$ where all the literals are assigned false. Hence, all three literal agents corresponding to three literals within $C_i$ should be included in $U$, and the clause agent $C_i$ should share a fraction of at least $1 - \epsilon$ with them. Therefore, the utility of the clause agent would be at most $1 - 1/\mu \cdot (4 \cdot (1 - \epsilon) - 3)^+ = 1 - 1/\mu \cdot (1 - 4\epsilon) < 0$ as $\mu < 1 - 4\epsilon$ and $\epsilon > 0$. This implies the utility of that clause agent is smaller than in the original data exchange $\boldsymbol{x}^0$ and causes the contradiction.

Therefore, when the input 3SAT instance is **NO** instance, we can conclude that the exchange $\boldsymbol{x}^0$ is core-stable, which concludes the coNP-hardness of determining whether $\boldsymbol{x}^0$ is core-stable in the constructed data exchange instance. $\qquad\square$

## C.3 Scarf's Lemma

In the main text of our paper, we show the data exchange game is balanced and then conclude the existence of core-stable data exchange. Next, we show that approximate core-stable data exchange can be constructed using the coalition matrix and the solution of Scarf's Lemma.

**Lemma 5** (Scarf's Lemma). *Let $n < m$ and $\mathbf{B} \in \mathbb{R}^{n \times m}$ such that the first $n$ columns of $\mathbf{B}$ form an identical matrix. Let $\mathbf{b}$ a non-negative vector in $\mathbb{R}^n_{\geq 0}$, such that the set $\{\mathbf{x} : \mathbf{B} \cdot \mathbf{x} = \mathbf{b}\}$ is bounded. Let $\mathbf{C}$ be an $n \times m$ matrix such that $c_{i,i} \leq c_{i,k} \leq c_{i,j}$ whenever $i, j \leq m, i \neq j$ and $k > n$. Then there exists a subset $O \subseteq [m]$ with size of $n$ such that*

- *$\mathbf{B} \cdot \boldsymbol{\delta} = \mathbf{b}$ for some $\boldsymbol{\delta} \in \mathbb{R}^m_{\geq 0}$ such that $\delta_j = 0$ for $j \notin O$, and*

- *For every $k \in [m]$, there exists $i \in [n]$ such that $c_{i,k} \leq c_{i,j}$ for all $j \in O$.*

Note that since we discretize the utility space in the construction of the coalition matrix, we are already $\epsilon$ away from a possible utility vector. Further, Claim 5 gives whether a particular utility vector is possible up to an $\epsilon'$ error. Therefore, we have all possible utility vectors up to $(\epsilon + \epsilon')$ error. By setting $\epsilon' = \epsilon$, we have the following proposition.

**Proposition 3.** *For any utility vector $\mathbf{v}$ that is attainable by a coalition $S$, there exists a utility vector $\mathbf{u}$ of the columns of $\mathbf{U}$ such that $u_i \geq v_i - 2\epsilon$ for any $i \in S$.*

Based on that, we can construct a core-stable data exchange by applying Scarf's Lemma.

**Lemma 6.** *For any $\epsilon > 0$, there exists a $\epsilon$-core-stable data exchange when the payoff functions are concave and the cost functions are convex.*

*Proof.* First construct the coalition matrix $\mathbf{C}$ and utility matrix $\mathbf{U}$ using parameter $\epsilon/2$ as described in Section 4.1. To meet the preconditions of Scarf's Lemma, we slightly modify $\mathbf{C}$ and $\mathbf{U}$ by considering all the singleton coalitions (where only one agent forms a coalition). Therefore, $\mathbf{C}$ is then changed to an augmented matrix with an identity matrix in the left part while a zero matrix is added to the left of $\mathbf{U}$. Similarly, we add slightly different $M$ to the blanks in the first $n$ columns of $\mathbf{U}$ like discussed before.

The two matrices then meet the preconditions of Scarf's lemma. We next construct a data exchange $\mathbf{x}$ according to the solution $(O, \boldsymbol{\delta})$ as follows: $\mathbf{x} = \delta_i \cdot \mathbf{x}^i$, where $\mathbf{x}^i$ is the data exchange archiving the $i$-th utility column of $\mathbf{U}$. According to the second guarantee of Scarf's Lemma and the concavity of the utility function, the deviation corresponding to every column of $\mathbf{U}$ cannot block $\mathbf{x}$. Therefore, by Proposition 3, we can then conclude that $\mathbf{x}$ is $\epsilon$-core-stable. $\square$

## C.4 Pivoting Algorithm

Next, we show the pseudo-codes of the pivoting algorithm for finding an $\epsilon$-core-stable exchange in Algorithm 2, which mainly follows the constructive proof of Scarf's Lemma [Sca67]. In line 2, we first construct the two matrices: coalition matrix $\mathbf{C}$ and utility matrix $\mathbf{U}$ with the input approximation parameter $\epsilon$. Then we apply the pivoting algorithm to find a solution of Scarf's Lemma, which further induces an $\epsilon$-core-stable data exchange. For completeness, we first introduce the concepts of *cardinal basis* and *ordinal basis*. Denote the size of the two matrices by $n \times m$, where $m$ is the number of possible coalitions.

**Definition 5** (Cardinal basis). Let $\mathbf{b} = \mathbf{1}$. Consider the polytope $P = \{\mathbf{C} \cdot x = \mathbf{b}\}$. A set of columns $B$ is a *cardinal basis* for $(\mathbf{C}, \mathbf{b})$ if (i) $|B| = n$ and; (ii) the submatrix induced by $B$ has a full rank.

**Definition 6** (Ordinal basis). A subset of columns $O$ is a called *ordinal basis* for the utility matrix $\mathbf{U}$ if (i) $|O| = n$ and; (ii) for every column $j \in [m]$, there exists a row $i \in [n]$ such that $U_i^O \geq U_{i,j}$, where $U_i^O = \min_{j \in O} U_{i,j}$.

During PA, it respectively maintains two bases $B$ and $O$ for each of the two matrices $\mathbf{C}$ and $\mathbf{U}$. These bases evolve until they are equal. To get to this, we will pivot to a new basis in each matrix. A pivot step in the coalition matrix $\mathbf{C}$ is defined like the usual linear programming pivoting step. A pivoting step in the utility matrix $\mathbf{U}$ is called an *ordinal pivot step*.

---

**Algorithm 2:** Pivoting algorithm for finding $\epsilon$-core-stable exchange

---

1   **Input**: Payoff functions $\{p_i\}_i$, cost functions $\{c_i\}_i$ and the parameter $\epsilon > 0$;

2   Construct $\mathbf{C}$ and $\mathbf{U}$ with parameter $\epsilon > 0$, payoff functions $\{u_i(\cdot)\}$ and cost functions $\{c_i(\cdot)\}$ ;

3   Update $\mathbf{C} \leftarrow (I_n \mid \mathbf{C})$ and $\mathbf{U} \leftarrow (\mathbf{0} \mid \mathbf{U})$ and then fill the blanks of $\mathbf{U}$ with slightly different sufficiently large value $M$. Let the size of $\mathbf{C}$ and $\mathbf{U}$ be $n \times m$;

4   Let $B \leftarrow \{1, 2, \ldots, n\}$ and $O \leftarrow \{1, \ldots, n\} \cup \operatorname{argmax}_{j \in [n+1, m]} U_{1,j}$;

5   **while** $B \neq O$ **do**

6      ▷ `Cardinal Pivot`

7      Let $j$ be the column in $O \setminus B$, $\mathbf{C}_B$ be the submatrix induced by $B$ and $\mathbf{b}_j$ be the column indexed by $j$;

8      Let $\mathbf{x}$ and $\mathbf{y}$ respectively be the solution of the linear equations $\mathbf{C}_B \cdot \mathbf{x} = \mathbf{1}$ and $\mathbf{C}_B \cdot \mathbf{y} = \mathbf{b}_j$;

9      Let $j^* \leftarrow \operatorname{argmin}_{j:y_j>0} \frac{x_j}{y_i}$;

10     Update $B$ by $B \leftarrow B \setminus \{j\} \cup \{j^*\}$ ;          `// update the cardinal basis`

11     **if** $B = O$ **then**

12       **break**;

13     ▷ `Ordinal Pivot`

14     Let $j_\ell$ be the column in $O \setminus B$ to be pivoted out;

15     Let $i_\ell$ be the row minimizer of column $j_\ell$, i.e., $U_{i_\ell, j_\ell} = \operatorname{argmin}_{j \in O} U_{i_\ell, j}$;

16     Let $j_r \leftarrow \operatorname{argmin}_{j \in O \setminus \{j_\ell\}} U_{i_\ell, j}$;

17     Let $i_r$ be the row minimizer of column $j_r$, i.e., $U_{i_r, j_r} = \operatorname{argmin}_{j \in O} U_{i_r, j}$;

18     Let $K \leftarrow \{k \in [m] \setminus O : U_{i,k} > U_i^{O \setminus \{j_\ell\}}, \text{for all } i \neq i_r\}$;

19     $j^* \leftarrow \operatorname{argmax}_{k \in K} C_{i_r, k}$;

20     Update the ordinal basis by $O \leftarrow O \cup \{j^*\} \setminus \{j_\ell\}$ ;     `// update the ordinal basis`

21   Find the solution $\boldsymbol{\delta}$ of the linear equation $\mathbf{C}_B \cdot \boldsymbol{\delta} = \mathbf{1}$;

22   Let $\boldsymbol{x} \leftarrow \sum_{i \in [m]} \delta_i \cdot \boldsymbol{x}^i$ with $\boldsymbol{x}^i$ corresponding the data exchange achieving utility vector $\boldsymbol{u}_i$ ;

23   **return** *the data exchange* $\boldsymbol{x}$;

---

**Definition 7** (Cardinal Pivot)**.** For the coalition matrix $\mathbf{C}$, given a basis $B = (j_1, \ldots, j_n)$ for the matrix consisting of $n$ of its columns, we can take any column $j \notin B$ and remove one of the columns from $B$ to get a new basis using standard linear algebra. Such a movement is called a *cardinal pivot*.

**Definition 8** (Ordinal Pivot)**.** For the utility matrix $\mathbf{U}$, given a basis $O = (j_1, \ldots, j_n)$ consisting of $n$ columns of the matrix, we can take any column in the basis and replace it with a unique column from outside the basis. Such a step is called an *ordinal pivot* step.

Once PA terminates, we get an identical basis $B$, which is both a cardinal basis and an ordinal basis. Then we solve the linear equation $\mathbf{C}_B \cdot \boldsymbol{\delta} = \mathbf{1}$ and get the weights for each column. Afterward, in Line 22, we calculate the weighted data exchange $\boldsymbol{x} = \sum_i \delta_i \cdot \boldsymbol{x}^i$ with $\boldsymbol{x}^i$ as the data exchange corresponding to the $i$-th column. The data exchange is guaranteed to be $\epsilon$-core-stable by Lemma 6.

