# OpenReview forum: "On the Existence and Complexity of Core-Stable Data Exchanges"
_NeurIPS.cc/2025/Conference — NeurIPS 2025 poster_

### Official Review · Reviewer_TXg6 · 2025-07-02

**Clarity:** 3
**Significance:** 3
**Originality:** 2
**Rating:** 4
**Confidence:** 3

**Summary:**

The paper considers a simple data-exchange scenario, in which every agent gains a payoff $p_i$ from getting data from another agent and looses a cost $c_i$ of sharing its own data, i.e. $u_i(x) = p_i(x_{-i}) -c_i(x_i)$, where $x_{-i}$ is the data bundle that $i$ receives from the exchange and $x_i$ is the one that $i$ gives. The authors study the classical notion of core stability in this context and show, in particular, that a core-stable data exchange always exists when the payoffs are concave and the costs convex. In the other cases, core-stable exchanges are not guaranteed to exist. They also show NP-hardness of assessing existence (when the sufficient conditions are not met) and PPAD-completeness of the problem of finding core-stable exchanges, even constraining the coalition size and under the sufficient conditions for existence and how one can use a pivoting algorithm (PA) to achieve approximate core stability. Lastly, they assess PA’s performance via experiments.

**Questions:**

Are there alternative concepts to core stability that are interesting to consider in this context?

**Ethical Concerns:**

["NO or VERY MINOR ethics concerns only"]

**Final Justification:**

I slightly raused my score based on the rebuttal phase. The authors addressed some of my concerns (especially the ones on the complexity results), in a convincing way.

**Limitations:**

yes

**Paper Formatting Concerns:**

No concerns.

**Quality:**

2

**Strengths And Weaknesses:**

**Positive points**:
The model explained in the paper is quite simple and general, as well as the conditions proved to be sufficient for the existence of a core-stable data exchange. The authors discuss how these conditions often hold in practice, which is a good point to state. I also appreciate the fact that, after showing hardness results, they propose a solution which seems to be usable in practice to approximate core-stability.

**Negative points**:
In general, I think that the paper shows a clever application of very classical and well studied notions in cooperative game theory and there is not much novelty in there from a theoretical perspective. I find the theoretical results per-se a bit weak: the only technically interesting one (at least to me) is the reduction to show PPAD-completeness. Moreover, it is well known that the concept of core stability is very hard to achieve in general: as much as they are still interesting, the hardness results are not surprising to me. The Scarf’s algorithm used in the last section of the paper, is known as well and not novel. Thus, if I have to judge this paper as a theoretical one, I think it is below the threshold for acceptance.

**Minor comments**:
Supergradident -> Supergradient on lines 92/93

---

> ### Author Rebuttal · Authors · 2025-07-31
>
> We thank the reviewer for the insightful comments. Here, we will address the questions and concerns respectively as follows.
>
> > **Weakness:** In general, I think that the paper shows a clever application of very classical and well studied notions in cooperative game theory and there is not much novelty in there from a theoretical perspective. I find the theoretical results per-se a bit weak: the only technically interesting one (at least to me) is the reduction to show PPAD-completeness. Moreover, it is well known that the concept of core stability is very hard to achieve in general: as much as they are still interesting, the hardness results are not surprising to me. The Scarf’s algorithm used in the last section of the paper is known as well and not novel. Thus, if I have to judge this paper as a theoretical one, I think it is below the threshold for acceptance.
>
> **Response:** While we respect the reviewer’s opinion, we remark that there are several problems where core-like guarantees are computable in polynomial time or the problem lies in a smaller complexity class.
> 1. In the exchange markets with rivalrous goods, a core-stable solution can be computed through a competitive equilibrium in polynomial time (under specific agent preferences),
> 2.  The top trading cycle algorithm (which runs in polynomial time) determines a core-stable solution in housing allocation,
> 3.   Further, since data is a non-rivalrous asset (agents are therefore not competing for this asset), the hardness results that exist in traditional economies need not generalize. For instance, the work of [ACGM] looks at computing core-stable allocations in settings where utility functions are just the accuracy functions and do not involve costs. In such a setting, core-stable allocations are trivial. Further, as [ACGM] prove in their paper, finding allocations that are simultaneously core-stable and fair is in the complexity class CLS (and thus unlikely to be PPAD-hard).
>
> Lastly, we would like to highlight that our hardness results continue to hold even when restricted to coalitions of constant size—a fact that we found somewhat surprising.
>
> > **Question:** Are there alternative concepts to core stability that are interesting to consider in this context?
>
> **Response:** Core-stability is usually the gold standard desiderata in co-operative economies, as it implies other guarantees like individual rationality, Pareto-optimality, etc. One could also consider relaxations of core-stability where deviating coalitions (depending on their size) need to achieve substantially larger utility, e.g., if a group of $k$ agents need to deviate from the grand coalition, then the deviating exchange must guarantee each of them obtains a utility that is strictly larger than $f(k)$ times their utility in the grand coalition, where $f(\cdot)$ is a decreasing function with $f(n) = 1$. Note that $f(k) = 1$ for all $k$ in standard core-stability. Such guarantees have been investigated in the context of public decision making in social choice [Fain et al. '18] and Federated learning [Chaudhury et al'22]. One may hope to get better computational guarantees in this relaxed version.
>
> [Chaudhury et al'22] Fairness in Federated Learning via Core-Stability. In NeurIPS'22.
>
> [Fain et al. '18] Fair Allocation of Indivisible Public Goods. In EC'18

---

> > ### Comment · Reviewer_TXg6 · 2025-08-07
> >
> > I thank the authors for their thoughtful response.
> >
> > I appreciate the clarifications and the detailed examples that you wrote. Probably I am just more accostumed to different settings that are usually harder w.r.t. core related problems.
> > The idea of relaxing the core as you explained is interesting as well (at least to me) and seems a nice futurr step to investigate.
> >
> > I am increasing my score.

---

> > > ### Author Response · Authors · 2025-08-07
> > >
> > > Thank you very much for your kind response and support on our paper!
> > > We appreciate your question about the extensions and are particularly encouraged that you find the idea of relaxing the core interesting — we agree it’s a promising direction. We will include it in the discussion section of our next revision.
> > >
> > > Thank you again for your constructive feedback!

---

### Official Review · Reviewer_f3Pk · 2025-07-02

**Clarity:** 3
**Significance:** 3
**Originality:** 3
**Rating:** 5
**Confidence:** 3

**Summary:**

The article considers a setting where agents can share data with one another in a federated manner, obtaining a payoff from the data they receive while incurring a cost for sharing their own. Data sharing is modeled as continuous, allowing for varying levels between no sharing and full sharing.

The authors focus on core-stable exchanges—vectors of exchange levels in which no coalition of agents can deviate and strictly improve the payoffs of all its members. By modeling the setting as a cooperative game, the authors study the existence and computational complexity of such exchanges, establishing sufficient conditions for their existence as well as hardness results for their computation.

To address these challenges, the authors propose an approximation scheme based on the Pivoting Algorithm, which is then empirically evaluated on a real-world dataset.

**Questions:**

Regarding the set of core-stable solutions, could you please comment on the following points:
1. Is the core-stable solution unique in general?
2. Which particular solution is approximated by the Pivoting Algorithm?
3. With respect to fairness—as mentioned at the end of the article—is there any chance that the Shapley value of your cooperative game lies in the core? If so, could it be approximated?

**Ethical Concerns:**

["NO or VERY MINOR ethics concerns only"]

**Final Justification:**

The paper is good. The setting is interesting and the article is quite rigorous. The authors replied to all my concerns. I've decided to keep my good score.

**Limitations:**

Yes

**Quality:**

3

**Strengths And Weaknesses:**

Strengths
- The article is well written.
- The proposed model is interesting and relevant.
- The paper is theoretically strong.
- The results are quite complete, including existence proofs, computational hardness results, and the proposal of an approximation scheme for the desired solution.

Weaknesses
- The empirical evaluation is limited, as the Pivoting Algorithm is tested on only one dataset and is not compared against any other approximation scheme.

---

> ### Author Rebuttal · Authors · 2025-07-31
>
> We thank the reviewer for the insightful comments. Here, we will address the questions and concerns respectively as follows.
>
>
> > **Weaknesses:** The empirical evaluation is limited, as the Pivoting Algorithm is tested on only one dataset and is not compared against any other approximation scheme.
>
> Response: We thank you for the suggestion on the evaluation. We have managed to add one more dataset from the latest official San Francisco road map [DataSF], which is continuously updated on a daily basis. This dataset includes 35,000 street intersections along with the corresponding source and destination roads for each intersection. We construct the graph by treating intersections as edges and the roads as vertices, and adopt the same parameter setting in our paper. Given that the graph is smaller and sparser than the road map of New York used in our paper, we have tested our method for 10 to 80 agents. We report the average statistics for the performance pivoting algorithm as follows:
>
> Time and size for the coalition matrix
>
>
>
> | n | 10 | 20 | 30 | 40 | 50 | 60 | 70 | 80 |
> | -------- | -------- | -------- |-------- |-------- |-------- |-------- |-------- |-------- |
> | Time (sec) |0.84| 1.56 |10.25 |15.78 | 22.81 | 61.15 | 71.18 |123.61|
> | Size | 1 | 10 |161 |239 |424 |913 | 1003 | 1366
>
>
>
> Time and iterations of the pivoting algorithm
>
> |n | 10 | 20 | 30 | 40 | 50 | 60 | 70 | 80 |
> | -------- | -------- | -------- |-------- |-------- |-------- |-------- |-------- |-------- |
> |Time (sec) | 0.01 | 0.03 | 0.04 | 3.38 | 9.24 | 79.33 |165.49 |205.14|
> |Iter | 1 | 12 |20 |156 |241 | 797 |1042 | 1154
>
>
> [DataSF] List-of-Streets-and-Intersections, Geographic-Locations-and-Boundaries, San Francisco Open Data.
>
>
> > **Question 1:** Is the core-stable solution unique in general?
>
>
> **Response:** In general, there can be multiple core-stable exchanges. Consider two agents, and let $x_{12}$ denote the data transferred from agent 1 to 2 and $x_{21}$ denote the data transferred from agent 2 to 1. Agent 1’s accuracy is given by $x_{21}$ and cost is given by $x_{12}$ (therefore net utility is $x_{21}-x_{12}$). Symmetrically, agent 2’s accuracy is given by $x_{12}$ and cost is given by $x_{21}$ (therefore net utility is $x_{12} - x_{21}$). One can easily observe that any exchange that sets $x_{12} = x_{21}$ is core-stable and there are several of them.
>
> > **Question 2:** Which particular solution is approximated by the Pivoting Algorithm?
>
> **Response:** In continuation of the answer to question 1, there can be multiple core-stable data exchanges, and since we do not impose any additional selection criteria, our algorithm can return any stable data exchange. Studying how to steer the algorithm toward desirable core-stable solutions—possibly via objective-guided pivots—is an interesting direction for future work. We also urge the reviewer to check our response to reviewer 5sUM, question 1, where we motivate payment mechanisms that can help us find core-stable data exchanges with additional desiderata.
>
>
> > **Question 3:** With respect to fairness—as mentioned at the end of the article—is there any chance that the Shapley value of your cooperative game lies in the core? If so, could it be approximated?
>
> **Response:** Shapley value is in the core for games with Transferable Utilities (TU). In TU games, a  single value $v(S)$ is assigned to each coalition $S$ of agents. The goal is to design a credit sharing rule, such that the grand coalition is core-stable, i.e., under this credit sharing rule, no coalition of agents can deviate and all get strictly better credit. In such a setting, Shapley value is in the core, i.e., choosing the credit sharing rule as the Shapley share ensures that the grand coalition is core-stable. Our setting is a Non-Transferable Utility (NTU) game, where one cannot freely transfer utility between players. In these settings, for each coalition, there are feasible data-exchanges that determine agent utilities.
>
> However, one could design a budget-balanced payment mechanism that can make our game behave closely to a TU game, albeit some drawbacks. Please see our answer to reviewer 5sUM, question 2 for a more detailed explanation.

---

> > ### Comment · Reviewer_f3Pk · 2025-08-04
> >
> > I thank the authors for their detailed responses and appreciate the additional empirical results. For a future version, the paper would benefit from experiments comparing PA against other approximation schemes..
> >
> > Regarding the Shapley value for NTU games, I was referring to value solutions as discussed in "McLean, R. P. (2002). Values of non-transferable utility games. Handbook of Game Theory with Economic Applications, 3, 2077-2120."
> >
> > I've decided to maintain my positive score.

---

> > > ### Author Response · Authors · 2025-08-09
> > >
> > > We sincerely thank the reviewer so much for the support and thoughtful feedback! We appreciate your suggestion regarding the experiments and aim to add a comparison between the pivoting algorithm and another approximation method in our future version.
> > >
> > > We also apologize for the delayed response regarding the Shapley value of NTU games. We find it most relevant to the problem studied in [ACGM], which also formulates the data exchange problem as an NTU game. In their setting, agents incur no cost for data sharing, and they have shown that there exists a core-stable and reciprocally fair (equivalent to the Shapley value) data exchange. We were considering whether their approach could be adapted to find a core with the Shapley value in our setting. While it cannot be directly applied, it remains a non-trivial and interesting question.
> > >
> > > In particular, their method considers zero-cost settings by iteratively improving toward the Shapley value while maintaining core stability, using the property that an exchange is core-stable if its exchange graph (agents as vertices, edge $(i,j)$ if $x_{i,j} < 1$) is acyclic. Unfortunately, this property fails with non-zero costs. For instance, if each agent incurs a super large cost when sharing more than $1$, i.e., $\sum_j x_{i, j} > 1$ for every $i\in [n]$, then even if everyone sharing everything yields no edges in the exchange graph, the data exchange is not core-stable because every single agent has the incentive to deviate.
> > >
> > > We believe it would be interesting to provide a more concrete characterization of core-stable data exchanges (e.g., generalizing the definition of exchange graph) under the sufficient conditions given by our paper. We also thank the reviewer for bringing it up. We will make sure to mention it as future work in our next revision.
> > >
> > > [ACGM] On the Theoretical Foundations of Data Exchange Economics, Hannaneh Akrami, Bhaskar Ray Chaudhury, Jugal Garg, Aniket Murhekar, EC’25

---

### Official Review · Reviewer_XfeZ · 2025-07-03

**Clarity:** 3
**Significance:** 2
**Originality:** 2
**Rating:** 4
**Confidence:** 3

**Summary:**

Data exchanges consider the agents sharing (not selling) data in a manner that is beneficial to all. Agents have utility in receiving data from others and cost in sharing their own data. This works studies the problem of finding core-stable solutions: one which is individually rational for all parties and another allocation does not that that all agents strictly prefer. They show that a core-stable solution exists if the payoff is concave and the cost convex, with counterexamples possible if either does not hold. Under an oracle model, it is shown that it is NP-Hard to determine if a core-stable allocation exists; under the sufficient stated conditions where such a solution does exist, it is PPAD to compute it. That said, a more practical algorithm is proposed to find epsilon-core stable allocations using a pivoting algorithm.

**Questions:**

See above.

**Ethical Concerns:**

["NO or VERY MINOR ethics concerns only"]

**Final Justification:**

The authors provided more context on the conceptual contributions of the work.

**Limitations:**

Yes

**Quality:**

3

**Strengths And Weaknesses:**

I found the paper to be very well written and clearly exposited. The technical model is clean and well justified and the concept of core-stability is natural in this setting. Further, the technical results are fairly complete insofar as a theory paper goes.

While I am not an expert in the area, my main concern is the overlap, both in conceptually contribution and originality to [ACGM, 2024 - On the Existence and Complexity of Core-Stable Data Exchanges]. This work, cited in the paper but not thoroughly discussed, also consider data exchanges that satisfy core stability. While the utility model is slightly different -- no explicit mention of cost and the utility is considered generally -- I don't see the utility function formulation here as being any more attractive. The conditions they require for core stability is that the utility be monotone: this is extremely natural and more general than here. They also give a local search algorithm that returns an epsilon-core stable allocation. Their work simultaneously also satisfies a fairness desiderata.

In light of this aforementioned work, I struggle to understand the contribution of this work. While I am convinced there's mathematical novelty here, at a conceptual level, what makes the results here more relevant/applicable to data exchanges than the prior work? What more is being said here that is of practical significance to data exchanges. At the very least, a fairly meaningful discussion about the relation to this prior work is necessary in my opinion given the significant overlap - there is very little of it in the work. The only mention is the earlier work does not consider cost (at the very end of the paper) which I am not sure is as important in data exchanges.

---

> ### Author Rebuttal · Authors · 2025-07-31
>
> We thank the reviewer for the insightful comments. Here, we will address the questions and concerns respectively as follows.
>
> > **Weakness 1:** While I am not an expert in the area, my main concern is the overlap, both in conceptually contribution and originality to [ACGM, 2024 - On the Existence and Complexity of Core-Stable Data Exchanges]. This work, cited in the paper but not thoroughly discussed, also consider data exchanges that satisfy core stability. While the utility model is slightly different -- no explicit mention of cost and the utility is considered generally -- I don't see the utility function formulation here as being any more attractive. The conditions they require for core stability is that the utility be monotone: this is extremely natural and more general than here. They also give a local search algorithm that returns an epsilon-core stable allocation. Their work simultaneously also satisfies a fairness desideratum.
>
> **Response:** We appreciate the reviewer’s comparison with [ACGM]. However, we emphasize that there are major important distinctions between our setting and that in [ACGM]. In particular, as the reviewer points out, [ACGM] does not account for costs, making its utilities monotone in the data exchange. It becomes trivial to obtain a core-stable data exchange in their setting – everyone exchanges all the data. This way, no agent can achieve a higher utility in any exchange. Therefore, ACGM investigates core stability in conjunction with another desideratum, i.e., fairness, and proves their existence results.
>
> In contrast, our work departs significantly from this assumption, by explicitly incorporating costs incurred by agents for the data sharing– a feature intrinsic to real-world data sharing (see next response, and also our answer to question 2 of reviewer 5sUM). In our model, an agent’s utility is defined as payoff minus cost, both of which are monotonic in data exchange, but in opposite directions. As a result, the overall utility is non-monotonic, which introduces significant complexity. In particular, [ACGM]’s results do not carry over, as we demonstrate in Example 1, a core-stable exchange may not even exist under the generality of ACGM’s model when such costs are taken into account. This motivates our investigation into sufficient conditions—such as concave payoffs and convex costs—that still capture important and realistic scenarios while allowing for meaningful analysis.
> Conceptually, our utility model aligns more closely with practical distributed learning frameworks such as federated learning, where agents must consider both the benefits and costs of participation. We appreciate the reviewer’s suggestion and will include a more detailed comparison with ACGM in the next revision of our paper to make these distinctions clearer.
>
> > **Weakness 2:** In light of this aforementioned work, I struggle to understand the contribution of this work. While I am convinced there's mathematical novelty here, at a conceptual level, what makes the results here more relevant/applicable to data exchanges than the prior work? What more is being said here that is of practical significance to data exchanges. At the very least, a fairly meaningful discussion about the relation to this prior work is necessary in my opinion given the significant overlap - there is very little of it in the work. The only mention is that the earlier work does not consider cost (at the very end of the paper) which I am not sure is as important in data exchanges.
>
> **Response:**  We believe that modelling costs incurred during data sharing is very fundamental and has already been extensively used in collaborative machine learning paradigms like federated learning [Murhekar et al. '23], [Murhekar et al. '25], [Karimireddy et al. '22]. All the foregoing work cite costs associated with acquisition (e.g., retailers offering reduced prices/ discounts/ vouchers to customers in exchange for their data, buying data from third party data collectors), storage, cleaning, and privacy (anonymizing data for privacy considerations and setting up physical and legal infrastructure for sharing it) in a data sharing. [Castro Fernandez. '23] argues that a proper cost-utility tradeoff analysis is required for any kind of data sharing.
>
> "Without a clear cost-benefit analysis, participants default in not sharing. As a consequence, many opportunities to create valuable data-sharing consortia never materialize, and the
> value of data remains locked."-- [Castro Fernandez. '23]
>
> Data exchange is an interesting economy under the umbrella of data sharing, where agents share data for mutual benefit (please also see our answer to reviewer 5sUM, question 2, where we elaborateon  the origin and the form of some of the cost functions in data economies).
>
>
> Incorporating costs makes the utility functions (payoff minus costs) non-monotone and significantly more complex to deal with. Despite this, we identify some natural sufficient conditions under which core-stable guarantees are achievable and also outline an algorithm that works well in practice.
>
>
> [Murhekar et al. '23]: Incentives in Federated Learning: Equilibria, Dynamics, and Mechanisms for Welfare Maximization, NeurIPS 2023
>
> [Murhekar et al. '25]: You Get What You Give: Reciprocally Fair Federated Learning, ICML 2025
>
> [Karimireddy et al. '22]: Mechanisms that Incentivize Data Sharing in Federated Learning, In Workshop on Federated Learning NeurIPS 2022.
>
> [Castro Fernandez. '23] Data-Sharing Markets: Model, Protocol, and Algorithms to Incentivize the Formation of Data-Sharing Consortia, In *SIGMOD'23*

---

> > ### Comment · Reviewer_XfeZ · 2025-08-08
> >
> > Thank you for your response. I have a better appreciation for the work now and have updated my score accordingly.

---

> > > ### Author Response · Authors · 2025-08-08
> > >
> > > We thank you very much for taking the time to reconsider our submission. We really appreciate your question regarding the comparison with [ACGM] and will add further discussion on the differences in setting and results in the related work section of our next revision.
> > >
> > > Thank you again for your support!

---

### Official Review · Reviewer_5sUM · 2025-07-04

**Clarity:** 3
**Significance:** 3
**Originality:** 3
**Rating:** 5
**Confidence:** 4

**Summary:**

The paper investigates the existence and computational complexity of core-stable data exchanges in collaborative data economies. It models data exchange as a cooperative game where agents balance payoffs from acquired data against sharing costs, aiming to prevent coalitions from deviating to more favorable local exchanges. The study establishes conditions for core-stability, analyzes its computational hardness, and show the practical performance of the pivoting algorithm.

**Questions:**

1. Is it possible to involve monetary transfer in the setting? Either between agents or involving enconomic issue related to the platform of data exchange (for example charging a small fraction of payoffs for each exchange).
2. Please explain more to justify the assumption of the convex cost functions.

**Ethical Concerns:**

["NO or VERY MINOR ethics concerns only"]

**Final Justification:**

The author addressed my concern on the assumptions and questions on settings.

**Quality:**

3

**Strengths And Weaknesses:**

Strengths:
1. It is novel to model the data exchange problem through cooperative game setting. The core stability is a good solution concept to evaluating the feasibility of data exchange platform.
2. Concrete theoretical results on the existence and complexity.

Weaknesses:
1. The assumptions (concave payoffs, convex costs) may not hold in real data exchange scenario. To be specific, for the concave payoffs, it is common that the payoff function w.r.t. acquired data is a step function, just as Example 1 illustrates. Intuitively, for instance, a learning model may not learn anything befere the size of training data is beyond some threshold. Notice that the author has provided several previous works on ML models with concave payoffs, I would not regard this as a big issue. As for the convexity of cost function, I cannot understand the illustration provided in Sec 1.1. Why do one share records in ascending order of costs involved for collecting the records? A direct way of sharing records is just sharing them in the order that they are collected, while it depends that collocting records requires increasing or decreasing marginal costs?

---

> ### Author Rebuttal · Authors · 2025-07-31
>
> We thank the reviewer for the insightful comments. Here, we will address the questions and concerns respectively as follows.
>
> > **Question 1:** Is it possible to involve monetary transfer in the setting? Either between agents or involving economic issue related to the platform of data exchange (for example charging a small fraction of payoffs for each exchange).
>
> **Response:** This is indeed an excellent question. While it is certainly possible to design payment rules tailored to specific applications, we believe that several important challenges must be carefully considered in the design of such rules.
>
> * **Budget Constraints:** In practice, agents often have limited budgets and may not be able to make or receive large payments as the mechanism dictates. This can violate the feasibility of the mechanism, even if it is theoretically sound.
> * **Mechanism Consistency Across Coalitions:** Core-stability guarantees with payments generally rely on the assumption that deviating coalitions continue to use the same payment rule. In decentralized and heterogeneous environments, there is no natural reason to expect this consistency, which may limit the robustness of such mechanisms.
>
> That said, we still believe that several interesting theoretical questions can be explored building upon our current framework—particularly in the context of computing core-stable data exchanges while simultaneously satisfying additional desiderata. We elaborate our points.
>
> We can introduce a payment rule for every exchange in a coalition. Let $\mathrm{pay}\_i(x,S)$ denote the payment that $i$ receives from the exchange $x$ in coalition $S$. Ideally, we would ensure that $\sum_{i \in S} \mathrm{pay}\_i(x,S) = 0$, implying that the mechanism is budget-balanced. The utility of each agent after payment is $a_i(x_{-i}, S) - c_i(x_i, S) + \mathrm{pay}_i(x,S)$. Two natural questions that follow are,
>
> 1. **Can we design a payment rule such that a core-stable data exchange can be computed in polynomial time?**
> 2. **Can we design payments that align with additional desiderata?** For instance payment mechanisms can be designed in order to award agents contributing “valuable data” (data that increases the total utilitarian welfare), e.g., $\mathrm{pay}\_i(x,S) = \psi_i(x, S) - (a_i(x_{-i}, S) - c_i(x_i)),$  where $\psi_i(x,S)$ denotes the contribution of agent $i$’s data to the total utilitarian welfare. This way each agent’s final utility in any contribution will be equal to $\psi_i(x,S)$. Under such payment mechanisms, does a core-stable data exchange exist? If it does, what is its computational complexity?
>
> If required, we can add this discussion to the next version of our paper.
>
>
> >**Question 2:** Please explain more to justify the assumption of the convex cost functions.
>
> **Response:**  A detailed view on the privacy cost and data acquisition can be found in Li and Raghunathan [Li and Raghunathan'14]: They adopt the financial compensation model from Laudon [Laudon'96]. Typically, data collectors (e.g. grocery store) collect (private) data from agents (eg. buyers) in return for financial compensation (e.g., a membership cards/ gift cards/ coupons of a grocery store). This compensation is often referred to as *sunk cost*. [Li and Raghunathan '14] (Section 3.1) describe two common methods through which the data collectors release the foregoing data to data buyers:
>
> * **Random Selection** (Results in Linear (trivially convex) cost functions): Here, the data buyer requests for data records of a particular type (specified by attributes/ sensitivity level etc.)-- essentially all the data records are homogeneous. Given a request of $k$ records, the data collector, randomly chooses $k$ data records of a particular type. This results in a cost function which is linear in the total number of records requested.
> * **Ordered Selection** (Results in Convex cost functions) : In contrast, if the data buyer does not impose constraints on record types—seeking broad or general-purpose information—the data collector may choose to release records in order of increasing individual cost or sensitivity. Early records are cheaper or less sensitive, while later records may carry greater privacy risks or legal implications. This ordering naturally leads to convex cost functions, reflecting increasing marginal cost as more data is released.
>
>
>
> Further, several prior works on incentive design and fairness in distributed learning systems, particularly in federated learning [Murhekar et al.'23], [Murhekar et al.'25], [Karimireddy et al.'22] adopt convex cost functions for modeling the agents’ costs.
>
> We also completely agree with the reviewer that an alternative way of sharing records is just sharing them in the order they are collected: However, the initial data points are easier or cheaper to collect (e.g., publicly available or high-frequency), whereas accessing additional, rarer, or more private data often incurs increasing marginal costs—due to privacy concerns, regulatory hurdles, and operational constraints, thereby again leading to convex cost functions.
>
> [Li and Raghunathan '14]:Pricing and disseminating customer data with privacy awareness. Decision support systems, 2014.
>
> [Laudon '96]: Markets and privacy. Communications of the ACM, 1996.
>
> [Murhekar et al. '23]: Incentives in Federated Learning: Equilibria, Dynamics, and Mechanisms for Welfare Maximization, NeurIPS 2023
>
> [Murhekar et al. '25]: You Get What You Give: Reciprocally Fair Federated Learning, ICML 2025
>
> [Karimireddy et al. '22]: Mechanisms that Incentivize Data Sharing in Federated Learning, In Workshop on Federated Learning NeurIPS 2022.

---

> > ### Comment · Reviewer_5sUM · 2025-08-01
> >
> > Thanks for your response. For the convex cost functions, the explanation is fine. Basically, I think it is a reasonable assumption, although not general enough. For the issue about involving monetary transfer, since it seems not a direct extension, maybe it would be another work with more concrete theoretical results.

---

> > > ### Author Response · Authors · 2025-08-08
> > >
> > > Thank you very much for your thoughtful question and valuable feedback. We agree — there are indeed many promising extensions beyond our current cost function assumption, as you mentioned. For example, subadditive cost functions. We will include a discussion on these extensions, as well as data exchange involving monetary transfers, as future directions in our next revision. Thanks again for your support!

---

### Decision · Program_Chairs · 2025-09-17

**Decision:**

Accept (poster)

**Comment:**

The submitted paper studies the stability of a data exchange protocol in which agents need to balance the benefits of acquiring data against the costs of sharing their own. Building on the concept of core-stability from cooperative game theory, the authors show that core-stable exchanges exist when agents have concave payoff functions and convex cost functions, common in domains like PAC learning. Relaxing these conditions can lead to instability. They also prove that finding a core-stable exchange is PPAD-hard. Furthermore, the authors provide a pivoting algorithm to find core-stable exchanges.

**Strengths**
* Theoretical contributions: The authors provide results on the existence of core-stable solutions under sensible conditions (real-world applicability is debateable) and demonstrate the computational hardness of determining such solutions.
* Practical relevance: The presented pivoting algorithm is a practical approach to approximate core-stable solutions, and its performance is evaluated on real-world datasets.
* Clarity and completeness: The paper is well-written, with a clear exposition of the model, theoretical results, and empirical evaluation.

**Weaknesses**
* Assumptions on payoffs and costs: The assumptions of concave payoffs and convex costs, while justified, may not hold in real-world data exchange scenarios. The authors provided detailed justifications in the rebuttal and are encouraged to include those in the revised version of the paper.
* Empirical evaluation: The empirical evaluation is limited, as the pivoting algorithm is tested on only two datasets and is not compared against other approximation schemes. A comparison with other approximation schemes would strengthen the paper.
* Overlap with prior work: One reviewer noted some overlap with [ACGM, 2024], which also studies core-stable data exchanges (they consider a special case of the submitted paper). The authors clarified the distinctions (e.g., incorporation of costs and non-monotonic utilities) but a more detailed discussion of related work would strengthen the paper.
* Theoretical novelty: One reviewer found some of the theoretical contributions somewhat incremental when viewed in the context of cooperative game theory but the authors elaborated in their rebuttal on the value of their results and how they add to existing ones. They should include this discussion in their updated paper.

**Discussion and rebuttal summary**
The authors actively engaged with the reviewers during the rebuttal phase and addressed all concerns in detail, resulting in an improved recommendation by two reviewers. The authors made an unanimous acceptance recommendation for significant contributions to the study of core-stable data exchanges, with a arguably stylized but still relevant model, rigorous theoretical results, and a practical algorithm for approximating solutions. While there are some limitations, the authors have addressed these concerns convincingly during the rebuttal phase. Hence I am following the reviewers’ recommendations and suggest the acceptance of the paper.